# “Sea Water” Supplemented with Calcium Phosphate and Magnesium Sulfate in a Long-Term Miller-Type Experiment Yields Sugars, Nucleic Acids Bases, Nucleosides, Lipids, Amino Acids, and Oligopeptides

**DOI:** 10.3390/life13020265

**Published:** 2023-01-18

**Authors:** Robert Root-Bernstein, Andrew G. Baker, Tyler Rhinesmith, Miah Turke, Jack Huber, Adam W. Brown

**Affiliations:** 1Department of Physiology, Michigan State University, East Lansing, MI 48824, USA; 2Department of Chemical Engineering and Biotechnology, University of Cambridge, Cambridge CB3 0AS, UK; 3MRC Laboratory of Molecular Biology, Francis Crick Avenue, Cambridge CB2 0QH, UK; 4Department of Chemistry, University of Chicago, Chicago, IL 60637, USA; 5Department of Art, Art History and Design, Michigan State University, East Lansing, MI 48824, USA

**Keywords:** prebiotic chemistry, “dirty experiments”, ATP, cAMP, amino acids, peptides, sugars, fatty acids, steroids, chemical ecology, sea water, minerals

## Abstract

The standard approach to exploring prebiotic chemistry is to use a small number of highly purified reactants and to attempt to optimize the conditions required to produce a particular end product. However, purified reactants do not exist in nature. We have previously proposed that what drives prebiotic evolution are complex chemical ecologies. Therefore, we have begun to explore what happens if one substitutes “sea water”, with its complex mix of minerals and salts, for distilled water in the classic Miller experiment. We have also adapted the apparatus to permit it to be regassed at regular intervals so as to maintain a relatively constant supply of methane, hydrogen, and ammonia. The “sea water” used in the experiments was created from Mediterranean Sea salt with the addition of calcium phosphate and magnesium sulfate. Tests included several types of mass spectrometry, an ATP-monitoring device capable of measuring femtomoles of ATP, and a high-sensitivity cAMP enzyme-linked immunoadsorption assay. As expected, amino acids appeared within a few days of the start of the experiment and accumulated thereafter. Sugars, including glucose and ribose, followed as did long-chain fatty acids (up to C_20_). At three-to-five weeks after starting the experiment, ATP was repeatedly detected. Thus, we have shown that it is possible to produce a “one-pot synthesis” of most of the key chemical prerequisites for living systems within weeks by mimicking more closely the complexity of real-world chemical ecologies.

## 1. Introduction

One of the general principles of science appears to be that order emerges from complexity within bounds set by system–level constraints [1,2]. Miller [3] demonstrated that given the input of energy in the form of heat and electrical discharges, even relatively simple systems composed of ammonia, methane, and hydrogen gases supplemented by water vapor could give rise to compounds of interest to the understanding of prebiotic chemistry and the origins of living systems. Following in Miller’s footsteps, most chemists have attempted to simplify or modify the original chemical conditions to better yield essential molecules of living systems, and with great success (e.g., [4,5,6]). However, in light of the order-from-complexity concept, one must wonder what would happen if Miller-type experiments were complexified rather than simplified. This possibility has led us to approach the question of prebiotic evolution as a problem in the evolution of complex chemical ecologies rather than the optimization of specific chemical pathways [2,7,8].

In consequence, we have chosen not only what Vincent et al. [9] have recently described as a “synthesized mixtures” approach over an “assembled mixtures of reagent grade chemicals” approach (see also [10]) to prebiotic experimentation but have begun to explore the effects of using what might be called “complex” or “dirty synthesized chemical mixtures” that more closely approach the messiness of real-world conditions [11,12]. Even “synthesized mixtures” generally originate from the use of reagent-grade gases and distilled, deionized water in most Miller-like experiments [3,6], but real atmospheres are much more complex, and water sources are never pure.

The purpose of these experiments was to explore whether novel products resulted from Miller-like experiments modified to last weeks to months by regassing the atmosphere and employing “dirty” chemical conditions resulting from the use of “sea water” salts and common geological minerals. The experiments were preliminary ones that employed some arbitrarily chosen conditions designed to explore possible avenues for future optimization and variation. The rationale for these experiments was that increasing the complexity of the environment in which the reactions take place might also increase the probability that novel reactions would take place, yielding compounds of importance to the origins of cellular life, such as nucleic acids, sugars, and fatty acids or lipids, as well as amino acids. Of course, it would not be possible to produce phosphate-containing compounds, such as cAMP or ATP, without a source of phosphates, such as hydroxyapatite, and the presence of magnesium was likely required in order to stabilize such phosphates as occurs in biotic systems [13]. Thus, we chose to supplement the “sea water” with soluble amounts of calcium phosphate and magnesium sulfate. The sulfate was also thought to be important for providing a possible source of sulfur for the synthesis of amino acids, such as cystine and methionine. We also speculated that the hydroxyapatite, magnesium sulfate, and/or various trace elements, such as iron and iron sulfate present in the “sea water” (or resulting from the presence of the sulfur in the magnesium sulfate), might function as such catalysts for sugar synthesis, as sugar synthesis in prebiotic conditions has previously been demonstrated in aqueous solutions only in the presence of mineral catalysts. Notably, Reid and Orgel were able to produce sugars in prebiotic conditions in the presence of hydroxyapatite [14], and other investigators have succeeded by using other catalysts, such as iron and titanium oxides (e.g., [15,16,17,18]). Amino acids are common products of Miller-type experiments, and we reasoned further that if alpha fatty acids, such as butyric acid, fumaric acid, and succinic acid, could be produced, these might participate in ester-mediated amide bond formation [19] while production of longer-chain lipids could catalyze peptide formation [20].

An additional feature of the experiments was to run the syntheses for five to eight weeks by regassing our apparatus and adding aliquots of “sea water” at weekly intervals so as to maintain a reasonably constant or increasing supply of reactants over extended periods of time. Such long-term Miller-type experiments have been relatively rare (reviewed in [4]), so it is not known what effect increasing concentrations of products, such as amino acids, might have on the probability of supporting polymerization into peptides or the emergence of other classes of compounds.

This paper reports the results of these preliminary experiments. As summarized in Figure 1, in addition to the amino acids produced in classic Miller-type experiments, we have evidence for the production of peptides, alpha acids, fatty acids, steroids, sugars, nucleic acid bases, and nucleosides.

## 2. Materials and Methods

### 2.1. Apparatus

A modified version of the original Urey–Miller apparatus (Figure 2) was constructed in a way that permitted regassing of the apparatus, integrated sample ports for ease of repeated sampling, and had electrodes to produce high-energy sparks. The overall design of the apparatus was the same as the Urey–Miller one, with a flask to hold water brought to a boil by means of a heating mantel, a 5-L flask with ports through which electrodes were placed, a refrigerated condenser unit to act as a heat sink, and a u-tube to collect the condensed material and feed it back into the flask. Three ports between the flask and the spherical gas element were provided with fittings controlled by valves that permitted the apparatus to be evacuated to near vacuum or filled with desired mixtures of ga to permit monitoring of the gas pressure/vacuum and to permit the “atmosphere” within the apparatus to be released under pressure. Two additional ports for sampling the liquid in the flask and u-tube were also provided and sealed with rubber sampling septa. The entire apparatus was fabricated from borosilicate glass (a point that is important in terms of the long-term use of the apparatus).

The electrodes were activated by a Marx generator powered by a variable voltage (up to 12 V) DC power supply with ten capacitors that yielded an output charge of approximately 250 kilovolts. The power supply was variable so that the frequency of discharge could be controlled, and it was set to produce sparks approximately every five to ten seconds.

All components of the apparatus that could contact the gases or water sources were rinsed with 70% ethanol followed by reverse-osmosis deionized, autoclaved water, and then sterilized in an autoclave at 140 °C under 1.5 psi pressure for 40 min before each experiment. The apparatus was tested for possible leaks by being evacuated using the vacuum pump until the pressure gauge read “zero,” and this reading was maintained for a minimum of 24 h. The water used in the experiments was also sterilized by autoclaving and reverse osmosis deionized. The absence of ATP was confirmed using the ATP assay system described below. Prior to the start of each experiment, a total of 350 mL of this water was introduced through the septa into the flask and u-tube of the apparatus using a sterile 100 mL syringe and non-coring sampling needle. The apparatus was then subjected three times to as complete a vacuum as possible (with the pressure gauge again reading “zero”), which was held for at least one hour each time in order to purge the water of as much gas (particularly oxygen) as possible.

### 2.2. “Sea Water”

Initially, as noted above, deionized, sterilized water was introduced into the flask and u-tube at the beginning of each experiment. Each time a 10 mL sample was taken from the flask or u-tube, it was replaced with an equal amount of “sea water” so that the concentration of salts and minerals slowly increased over time. The rationale for this procedure was based on our ignorance of the concentrations of salts that might have been present in diverse types of water across the globe and the fear that too high a concentration of some elements or minerals might poison prebiotic reactions. Slowly ramping up the salt/mineral concentrations permitted us to be able to observe a range of conditions with any single experiment. While not optimal in terms of controlling the conditions of the experiments, it was, nonetheless, easily possible to monitor the concentration of phosphates as a general measure of the increases in other mineral concentrations, and the procedure could be reproduced from one experiment to the next or varied, as results might suggest.

“Sea water” was created by dissolving 35.5 g Alessi Sea Salt (evaporated from the Mediterranean Sea) in 2.0 L of deionized, sterilized water, creating a concentration of salts and minerals one-half that of normal sea water. The rationale for using this decreased concentration of salts was the assumption that the concentrations of salts in ocean waters were less than at present because the leaching of minerals from geological deposits and run-offs from rivers had been going on for far less time. This solution was augmented with 1.0 g of hydroxyapatite [Ca_10_(PO_4_)_6_(OH)_2_] (Sigma Aldrich, St. Louis, MO, USA) and 1.0 g magnesium sulfate heptahydrate (Epsomite or Epsom salt [MgSO_4_·7H_2_O]) (Sigma Aldrich). These two minerals were chosen as a starting point for our experiments because they are both abundant worldwide and would have been present in many locations in which prebiotic chemistries were taking place. The concentrations of hydroxyapatite and magnesium sulfate were chosen conservatively to ensure their solubility and were, otherwise, arbitrary. The resulting solution was tested during every experiment using the mass spectrometry and ATP testing detailed below to ensure that it contained no detectable biological molecules.

As a consequence of the supplemented “sea water” being added in small aliquots over time to the reaction mixture, it is likely that the concentrations of minerals represent a lower bound of what may have been present during many prebiotic chemical environments and may have reached far higher concentrations at least in some locations on Earth. As noted in the Introduction, these were designed as preliminary experiments to explore possible avenues for future optimization and variation and not as attempts to model any particular environment or location in the prebiotic Earth.

### 2.3. Regassing

After the water had been degassed, the gases comprising the “atmosphere” within the apparatus were added by means of a gas manifold that permitted each gas to be introduced in a controlled manner. The gas proportions were 40% ammonia, 40% methane, and 20% hydrogen, the total producing 1 atmosphere of pressure and mimicking the original conditions employed by Miller [3]. Although there is much controversy over the make-up of the prebiotic Earth atmosphere, we chose to stick with Miller’s “recipe” in order to limit the number of novel variables introduced into the experiments.

The apparatus was regassed approximately every seven days using the following procedure: the valve to the vacuum pump was opened, the vacuum pump turned on, and the gas inside the apparatus was evacuated until the liquid inside began to bubble. This process retained most of the gases dissolved in the liquid and did not yield a complete vacuum but did remove most of the existing “atmosphere,” resulting in about a 90% vacuum (0.1-atmosphere pressure). The vacuum was then shut off, the gas line switched on, and ammonia, methane, and hydrogen were added proportionally, as above, to return the 0.1-atmosphere “vacuum” to 1.0-atmosphere pressure.

### 2.4. Sampling

The heater was turned off for at least an hour prior to sampling so as to permit the liquid in the flask and the u-tube to cool. Then, 5 mL samples were obtained from the flask and the u-tube via their sampling ports using a sterile technique (sterile gloves were donned, rinsed with 70% ethanol, and dried before the apparatus was touched) employing sterile, individually wrapped syringes, and autoclaved, non-coring sampling needles. A sterile technique was then used to replace the 10 mL of liquid withdrawn with 10 mL of augmented “sea salt”, keeping the volume of water in the apparatus constant throughout the duration of the experiment and slowly raising the concentration of salts and minerals in solution.

ATP, pH, and phosphate tests were carried out immediately following sampling (see below). Samples for mass spectrometry were placed in RNA-free Nunc vials, sealed, and, if not immediately prepared, then refrigerated (for no more than two days) prior to preparation. Some samples were also frozen for future use (though none of the results reported here involve such frozen samples).

### 2.5. Ultraviolet (UV) Spectroscopy

An initial set of experiments were run to determine whether regassing increased the concentrations of compounds produced. During this initial set of experiments, and only in these, a constant concentration of “sea water” at 1/10 dilution was used, and no additional “sea water” was added following each sampling. Then, 100 µL samples were extracted from the flask portion of the apparatus every few days, and the UV spectrum from 190 to 340 nm was obtained on a SpectraMaxPlus scanning UV-Visible light spectrometer. Samples were subjected to spectroscopy immediately following their removal from the apparatus, and the resulting absorption curves were then plotted to provide a rough estimate of changes in the total concentration of compounds produced as a function of time.

### 2.6. Mass Spectrometry

Samples were prepared for gas-chromatogram/mass spectrometry (GC/MS) by trimethylsilylation using N-Methyl-N-trimethylsilyl-trifluoroacetamide (MSTFA) (Sigma Aldrich P/N 394866-10X1ML). Then, 100 µL samples were placed in 2 mL amber autosampler vials with micro-inserts and evaporated to dryness using vacuum centrifugation (General Electric high capacity, thermally-protected vacuum pump) in a Savant Speed Vac Concentrator. The samples were then redissolved in 140 µL of a mixture of MSTFA and pyridine (Sigma Aldrich ACS reagent, ≥99.0%, Product #360570), three to four by volume, sealed with vial caps, and placed in a sealed, light-opaque container with desiccant. The reaction was permitted to proceed overnight (at least 12 h) at room temperature. After the reaction was complete, the liquid portion of each sample was transferred to glass inserts, leaving any solid material behind, and the inserts were placed back into the mass spectrometry vials from which they came. The vials were recapped, placed back into the desiccant container, and the samples were analyzed immediately thereafter using an Agilent A Gas Chromatograph-Mass Spectrometer with an Agilent CP9013 J&W VF-5ms GC Column, 30 m, 0.25 mm, 0.25 µm, 10 m EZ-Guard, 7-inch cage. The GC–MS was tuned using facility-supplied standards as well as experiment-appropriate standards, such as urea (PHR1406-1G, Pharmaceutical Secondary Standard), glutamic acid (G0355000, European Pharmacopoeia (EP) Reference Standard), lactic acid (PHR1215 Pharmaceutical Secondary Standard; Certified Reference Material), alanine (A0325000 European Pharmacopoeia (EP) Reference Standard), glycine (G7126 ReagentPlus^®^, ≥99% (HPLC)), serine (S0450000 European Pharmacopoeia (EP) Reference Standard), asparagine (Y0000305 European Pharmacopoeia (EP) Reference Standard), and/or aspartic acid (A1330000 European Pharmacopoeia (EP) Reference Standard), all from Sigma-Aldrich (St. Louis, MO, USA).

Identification of products was performed using Agilent Chemstation software version LTS 01.11 to visualize mass chromatograms and produce mass spectra for matching to the NIST database (versions 11.1 through 20.0). The quality of match (QOM) of each spectrum was aggregated (see Appendix A, which reports on all compound identifications above QOM of 50). In general, a QOM of 90 (out of a possible perfect match of 100) was utilized as a criterion for inclusion in the data reported below; however, for compounds of particular interest to origins of life chemistry, such as some amino acids, oligopeptides, sugars, and nucleic acid-related compounds, QOM as low as 50 percent were reported, with the clear understanding that these matches were highly questionable and in need of further investigation in the future. In most of the cases in which low QOM are reported, either the compound is related to another compound identified with a better QOM or other experimental methods suggest the presence of the compound.

### 2.7. ATP Assay (Luciferin/Luciferase)

The presence of ATP was determined via an AccuPoint^®^ Advanced ATP Reader (Neogen Corporation, Lansing, MI, USA, Item No. 9903), which uses a photomultiplier system to sense photon production in a luciferin/luciferase system in the presence of adenosine triphosphate (ATP). Samples were tested by applying c. 10 µL to AccuPoint^®^ Advanced Samplers, Water (Neogen Corporation, Lansing, MI, USA, Item No. 9906), which have a sensitivity of 10 fmole/sample (10 relative light units equals 1 fmole). All positive values were confirmed by multiple measurements (between 2 and 6 trials) and accepted only if control materials (deionized, sterilized water, and the original “sea water” sources) tested negative for ATP on an equivalent number of tests. Results were interpreted in relation to a standard curve of known ATP concentrations.

### 2.8. cAMP Assay (Enzyme-Linked Immunoadsorption Assay—ELISA)

A high-sensitivity competitive ELISA assay was used to measure cyclic adenine mononucleotide (cAMP) (ENZO Life Sciences, ADI-900-067A, sensitive to 0.027 pmol/mL). Assays were carried out according to the manufacturer’s instructions and plotted against a standard curve of concentrations of the appropriate mononucleotide.

### 2.9. Phosphate Assay

Approximate phosphate concentrations were determined using Bartovation Phosphorus and Phosphate Detection Test Strips (Part # PWQ09V50), sensitive to 0–100 ppm (Bartovation, Queens, New York, NY, USA).

### 2.10. pH Measurements

Approximate pH was determined using UltraCruz^®^ pH Indicator Strips (Catalogue # sc-3667), Santa Cruz Biotechnology, Inc., Dallas, TX, USA.

## 3. Results

### 3.1. Increased Yields of Compounds through Regassing

The first experiment performed was simply to determine whether the use of the regassing procedure yielded greater concentrations of products over time. The concentration of “sea water” was kept constant in this particular set of experiments so as to limit the number of variables. The increase in product concentration was confirmed via UV spectroscopy (Figure 3). If the absorbance is approximately determined by the concentration of compounds in the solution, then running the apparatus after regassing for three weeks resulted in about four times the concentration of products were found after about one week. Later experiments were extended to five or six weeks with regassing each week, resulting in proportionally greater concentrations of products. During this initial experiment, no attempt was made at this time to identify the molecules produced, although the peak around 310 nm is suggestive of aromatic compounds such as tryptophan, nucleic acid bases, and/or steroids, which were later identified by mass spectrometry (see below).

### 3.2. Initial Compound Identification

The second set of experiments was designed to explore the effects of adding “sea water” supplemented with calcium phosphate (hydroxyapatite) and magnesium sulfate (Epsom salt, Epsomite). As per the Methods, samples were tested weekly for pH, the concentration of phosphates, ATP, and two sets of cAMP and cGMP; ELISA tests were carried out. All samples were subjected to GC–MS in order to identify the synthesis of other compounds. Notably, the pH of the flask and u-tube portions of the apparatus each started out with the same pH, which was about 9.5–10.0, and while the u-tube pH remained at this value throughout the experiments, the flask pH generally fell into the range of 6.5–7.5, presumably due to the accumulation of amphiphilic compounds. Due to the procedure used for adding the supplemented “sea water” in 5 mL aliquots to replace samples, the phosphate level rose from undetectable to 100 ppm over a period of five weeks and reached 200 ppm at eight weeks. No measure of the magnesium concentration was carried out, but given that the “sea water” was augmented with the same concentration of magnesium sulfate as calcium phosphate, it can be assumed that the magnesium concentration was in the same range as the phosphate concentration.

As expected, amino acids appeared within a few days of beginning each experiment. Confirmation that sample preparation was properly carried out was provided by using control samples of key compounds, such as glycine, alanine, asparagine, serine, and urea, and comparing the spectra of the pure controls with those of the compounds identified in the experimental preparations (e.g., Figure 4).

Among the first compounds to appear in these experiments were what might be called “reactant” molecules, from which many other compounds could be synthesized, which were found during the first week of all the experiments, and these included urea, formamide, acetic acid or acetamide, oxalic acid, and glycerol (Figure 5 and Table 1). Figure 6 and Figure 7 provide GC–MS spectra, identifying these products from the reaction mixtures. Additional spectra are shown in Appendix B.

### 3.3. Amino Acids

“Reactant” molecules were always accompanied during the first week of the experiments by amino acids, most commonly glycine, alanine, serine, aspartic acid, leucine, and tryptophan, although most of the biotic amino acids were found repeatedly by the end of four-to-six week experiments (Figure 8, Figure 9, Figure 10 and Figure 11 and Table 2). Probably due to the presence of magnesium sulfate in the “sea water”, both cysteine, methionine, and their metabolites were also repeatedly observed by GC–MS (Table 2 and Appendix C). We assume that these amino acids were racemic mixtures since there is no chiral selection mechanism present in our experiments, and the mass spectrometry methods do not distinguish L- from D- amino acids. Additional spectra are in Appendix C.

Additionally, beginning in the fifth week of the experiments, dipeptides began to appear, and alanyl–glycine (Figure 12), alanyl–alanine, leucyl–alanine, and glycyl–glutamic acid were observed in several independent experiments (Table 3). The tripeptide alanyl–alanyl–alanine was also observed in one experiment after the fifth week, although the quality of the identification by mass spectrometry was low (Table 3).

### 3.4. Sugars

In addition to amino acids, a wide range of monosaccharides were produced in the experiments (Figure 13 and Table 4), including fructose, glucose, mannose, ribose, and many of their metabolites, such as levo-glucosan, gluconic acid, ribonic acid, ribo-hexos-3-ulose, and ribono-1,4-lactone (Figure 14 and Figure 15 and Appendix D). Two disaccharides, maltose and trehalose, were repeatedly observed, though the identification of trehalose was consistently of poor quality and, therefore, questionable (Table 5).

### 3.5. Nucleic Acid Bases, Nucleosides, and Nucleotides

GC–MS also identified nucleic acid bases repeatedly in the experiments, including purine metabolites, adenosine, guanosine, and dihydrouracil (though the quality of the match for guanosine and uracil was marginal) (Figure 16 and Table 6). We assume that we made not only D-adenosine and D-guanosine but L-forms as well, but mass spectrometry does not differentiate these. Figure 17 and Figure 18 and Appendix E illustrate some of the identified compounds listed in the table.

Because we found that mass spectrometry (both GC and liquid chromatography–MS) was unable to detect concentrations of nucleic acids below 10 nM, additional non-spectroscopic methods were employed. The presence of nucleic acids was further verified by ELISA, which demonstrated increasing concentrations of cyclic adenosine monophosphate (cAMP) reaching more than 1 pmol/mL (Figure 19) during each of the two separate experiments. The cAMP results replicated for each experiment were repeatedly negative for the original “sea water” solution. The reliability of the cAMP data was further confirmed using a luciferin–luciferase reaction to demonstrate the presence of adenosine triphosphate (ATP) in four separate experiments (Figure 20) beginning as early as week four of the experiments but most often beginning to appear in week five when the phosphate (and presumably magnesium) concentration had reached 100 ppm. Once again, however, it is important to note that ATP appeared in only four of eight experiments that ran for six weeks or more, and the reasons for the failure to produce ATP in four of these experiments are unknown but may involve undetected oxygen leaks or simply effects of variations in initial conditions by altering the synthetic pathways to ribose and the nucleic acid bases, and/or by oxidizing the ATP itself as quickly as it is formed. The possibility of contamination is extremely unlikely for reasons that will be addressed below.

### 3.6. Fatty Acids and Steroids

In addition to amino acids, sugars, and nucleic acids, the GC/MS experiments also repeatedly yielded a range of fatty acids ranging from C4 through C20, as well as a few steroids (but never cholesterol or phospholipids, a point that will be amplified in the Discussion section) (Figure 21 and Table 7). The GC–MS spectra and NIST identifications of some of these compounds are illustrated in Figure 22, Figure 23 and Figure 24 and in Appendix F.

Figure 25 illustrates a GC–MS chromatogram for the end of week three of the synthetic process, after two regassings of the apparatus and with phosphate and magnesium concentrations of approximately 200 ppm. The chromatogram has been annotated with the NIST identifications of the compounds present in each peak and clearly shows the presence of amino acids, sugars, fatty acids, as well as the nucleic acid base adenine and adenosine.

### 3.7. Controls for Possible Sources of Contamination

Long-term experiments with repeated sampling, such as the ones reported here, are potentially subject to contamination. We, therefore, emphasize that certain types of molecules that would have been expected to be present if extremophile bacteria or human contact contaminated our experiments were never identified in any experiment.

We begin with possible bacterial contamination. Because of the conditions within the apparatus (anaerobic with a continuous flask temperature of 100 °C), only select extremophile bacteria could possibly have become contaminants. However, many types of compounds that such extremophiles produce were never observed. These absent compounds included the amino acids arginine, glutamine, and tyrosine. Common bacterial peptides, such as the cell–wall component muramyl dipeptide and the tripeptide glutathione, also were not observed. No polysaccharides or starches were identified. The most common extremophile bacterial lipids are tetra-ether lipids, glycerol-ester lipids, and phospholipids that characterize lipid membranes [21,22], none of which were identified in any experiment. Nor were the nucleic acids, cytosine, or thymine ever found. Thus, although we did not test directly for the presence of such extremophile organisms, using, for example, a polymerase chain reaction assay for mitochondrial genes, the absence of so many key molecules from our results provides a strong case against their presence and, therefore, their participation in producing the molecules that we did identify.

Human contamination is also unlikely due to the strict use of sterile techniques at all times while handling, loading, and sampling the apparatus. The effectiveness of our sterile technique was evident in the absence of peptides, such as glutathione, and disaccharides, such as sucrose, lactose, and maltose. Phospholipids, glycerophspholipids, phosphotidylcholine, and cholesterol, which are the most abundant lipids present in finger grease [23], were never observed in any experiment. It is further unlikely that the repeated pattern of finding ATP about four or five weeks into the experiments was due to human error contaminating the ATP test solely around that time in each of the four experiments. Furthermore, if our observation of adenosine was the result of contamination, then cytosine, uridine, and inosine should also have been observed, but they were not. Finally, contamination during sample processing for GC–MS was extremely unlikely as no contaminant compound was ever observed in the spectra of the control compounds (urea, amino acids, etc., e.g., Figure 4) that were run at the beginning of every spectrometry series.

One final type of evidence also argues against human or microbial contamination as a cause of our results and particularly addressed whether the abiotic synthesis of sugars and nucleic acids occurred during the experiments. We noted in our Methods that our apparatus was constructed from borosilicate glass, as it must be in order to resist the high heat conditions necessary for its operation and autoclaving. However, after several years of running the apparatus, we noticed that our mass spectrometry samples were increasingly contaminated with boric acid (Figure 26), an observation that, after eliminating all other possible sources of boron from components of the apparatus, such as the rubber septa, the “sea water” and added minerals, and the reagents used for preparing mass spectrometry samples, we attributed it to a very slow deterioration of the glass caused by long periods of time exposed to ammonia at high heat. Two striking modifications in the products of the experiments accompanied this boric acid contamination. One was the production of boronate sugars (Figure 27 and Figure 28), which could not possibly have originated from human contamination. The other modification was the loss of adenine and adenosine (Table 6 and Figure 18) from the samples and the appearance, instead of uracil, dihydrouracil, and azauracil (Appendix G). The observation that samples contained either adenine-related compounds or uracil-related compounds, but never both (and never cytosine- or thymidine-related compounds), argues, firstly, against contamination as a source of these nucleic acids and, secondly, for the possibility that boron-containing minerals may play a role in the catalysis of uracil-related compounds.

To summarize, the results of the experiments using “dirty” water and repeated regassing of the Miller-style apparatus yielded greater concentrations of compounds and a wider range of products than the original Miller experiment. In particular, sugars, amino acids, some peptides, several nucleic acids, alpha acids, and fatty acids were produced. The typical time course of the appearance of these compounds is summarized in Table 8. Notably, except for some of the sugars, the rest of the compounds only began to appear after regassing at least twice, which may explain why some of these compounds have not been observed in previous, shorter-term studies.

## 4. Discussion

In practice, the experiments reliably produced well-studied precursors to many prebiotic molecules, such as urea, formamide and glycerol, most (though not all) of the biotic and some abiotic amino acids, the key biotic sugars, a range of fatty acids and steroids, and, in some experiments, evidence of nucleic acids, including adenosine, cAMP, and ATP. Among the amino acids were two that were not found in Miller’s original experiments [3], which were cysteine and methionine. These amino acids were later produced by Miller through the addition of H_2_S in the gases [6,23], but in this case, they presumably resulted from the presence of a source of sulfur from the magnesium sulfate. Notably, by regassing the apparatus and running the experiments for many weeks, the yield of total products increased (Figure 2), resulting in the presence of sufficient amino acids and their derivatives to permit polymerization into di- and tri-peptides. Similarly, the production of key sugars, including ribose, in an environment including phosphates and nucleic acid bases, appears to have made possible the production of cAMP, ATP, and, possibly, guanosine. Whether the presence of magnesium permitted their stabilization will require additional research, and, if it does, more experiments will be needed to determine the concentrations of phosphates and magnesium required to optimize nucleoside production and stability. The observation that boron appears to alter the types of nucleic acid bases produced may also be a clue of importance for understanding what minerals are needed to catalyze the range of products characterizing living systems and what minerals might have “poisoned” essential reactions. Notably, borate compounds have been found to stabilize nucleic acids [24], making the results reported here concerning the shift from adenine-like compounds to uracil-like ones possibly significant.

As would be expected, precursors molecules (amino acids, monosaccharides, nucleic acid bases, short-chain fatty acids) appear a week or two prior to the more complex molecules (peptides, disaccharides, nucleotides, longer-chain fatty acids) (Figure 29 and Table 8). We emphasize that each one of these classes of compounds has independently been synthesized previously in Miller-type experiments, for example, amino acids [3,6,24], peptides [25], nucleic acid bases [26,27,28], sugars [14,15,16,17,18], and fatty acids [29,30]. In general, these previous Miller-type experiments have previously been explored as means to generate one particular class of compounds, such as amino acids or nucleic acids, and conditions have generally been optimized for the production of that particular compound class. Our experiment differs in having no particular class of compounds as a goal and functioning as an exploratory rather than an optimizing exercise. We must point out, however, that Saladino, et al. [31] have reported the synthesis of amino acids, lipids (including arachidonic and eicosatrienoic acids), and nucleic acid bases (but not nucleosides or sugars), starting with pure formamide solution heated in the presence of meteorite particles so that it is known that several classes of prebiotic compounds can be synthesized under Miller-like conditions with an appropriate mix of mineral catalysts present. It does not appear that anyone has previously demonstrated the production of amino acids, peptides, fatty acids, sugars, nucleic acids, and nucleosides under one set of conditions.

In short, by increasing the time that the experiments ran with a reasonably consistent atmosphere and by adding sea salts augmented with calcium phosphate and magnesium sulfate, we believe that we have produced the conditions necessary for a one-pot synthesis of all major classes of compounds required for the origins of living systems. Better optimized conditions may help to define the environments, in which living systems are most likely, or least likely, to have originated. In this context, it is likely that the addition of clays or minerals, other than calcium phosphate or magnesium sulfate, will alter the distribution and types of products. Iron and titanium oxides and ferrous sulfate, are obvious examples of minerals that should be explored since iron oxides have been employed as catalysts for sugar production in prebiotic experiments [14,15,16,17,18], and iron–sulfur clusters can coordinate with and be stabilized by cysteine-containing peptides and mediate the assembly of iron–sulfur cluster peptide complexes that can drive enzymatic reactions [32]. Iron complexes have also been implicated in the production and breakdown of universal metabolic precursor compounds [31]. A listing of the many other mineral catalysts of prebiotic reactions would be too long to include here, but some representative publications include [33,34,35,36], and a brief review can be found in [37]. More complex atmospheres (including CO_2_, CO, H_2_S, NO, etc.) may also increase or significantly alter the molecular complexity, types, or ratios, of products. Such experiments could also be designed to more closely mimic non-Earth environments, in which prebiotic chemistries can take place [38,39,40,41,42]. Exploring combinations of these various atmospheres with sets of minerals might more accurately mimic real-world environments in which prebiotic chemistries occurred. Finally, the production of nucleic acid precursors along with amino acids opens up new possible pathways within complex Miller-like experiments for performing the type of amino-acid-adenylate-mediated peptide syntheses carried out in purer experimental conditions [43,44].

## 5. Conclusions

We report a possible advance in mimicking real-world environments through our “dirty” prebiotic experiments and possible “one-pot syntheses” of all the essential classes of molecules required for the emergence of living systems. We are, however, fully aware of the severe limitations of the present study, many of which can be addressed by further experimentation. We do not have data on amounts of each compound formed in each sample, though some sense of relative concentrations can be gained from the peak heights in the original chromatograms (Figure 25 and Figure 26). One difficulty is that compound spectra overlap, creating serious difficulties in determining how much of each observed peak is due to the contribution of any particular compound. Additionally, for each compound identified, of which there are many, titration spectra based on the pure compound will be needed with which to compare the peak heights derived from the chromatograms; this will involve a great deal of additional work. The use of the atmosphere (ammonia, methane, hydrogen) originally employed by Miller [3] is certainly questionable given subsequent research on the atmosphere of the primordial earth (e.g., [38,39,40], nor is it representative of many other planetary atmospheres (e.g., [41,42], so that other mixtures of gases should be explored using the “dirty” approach employed here. Similarly, the use of Mediterranean sea salt at low concentrations relative to those present in modern sea waters is similarly open to modification in order to mimic freshwater, sea, lake, and pond waters, as well as hydrothermal vents, hot springs, etc. Many other minerals, other than or in addition to calcium phosphate and magnesium sulfate, should be explored to augment “dirty” conditions (e.g, [32,33,34,35,36]). We must rethink the assumption that the borosilicate glass used as part of the experimental apparatus is entirely non-reactive and consider instead that it may play an essential role in the synthetic environment as a reactive surface. Much longer experiments should also be performed to determine whether it is possible to produce more and longer peptides, di- and polysaccharides, di- and polynucleic acids, etc. We also believe that additional energy sources, such as ultraviolet light cycles, freeze–thaw cycles, wet–dry cycles, etc., should be introduced into these “dirty” experiments not only to drive novel reactions but also to act as selective agents to limit the chemical combinatorial explosion that can, otherwise, be expected to result [37].

In sum, it appears that a possible way to evolve living systems is first to evolve chemical ecosystems complex enough to support the range of necessary chemical reactions. Increasingly “dirty” experiments are, however, only a first step in that direction. Prebiotic evolution will also require increasingly complex environmental selection pressures, such as light–dark, wet–dry, and freeze–thaw cycles, in order to control the “explosion” of chemical species that complex environments will engender [36]. Real progress will occur when we can model both the chemical complexity of prebiotic environments and also their range of physicochemical selection pressures.

## Figures and Tables

**Figure 1 life-13-00265-f001:**
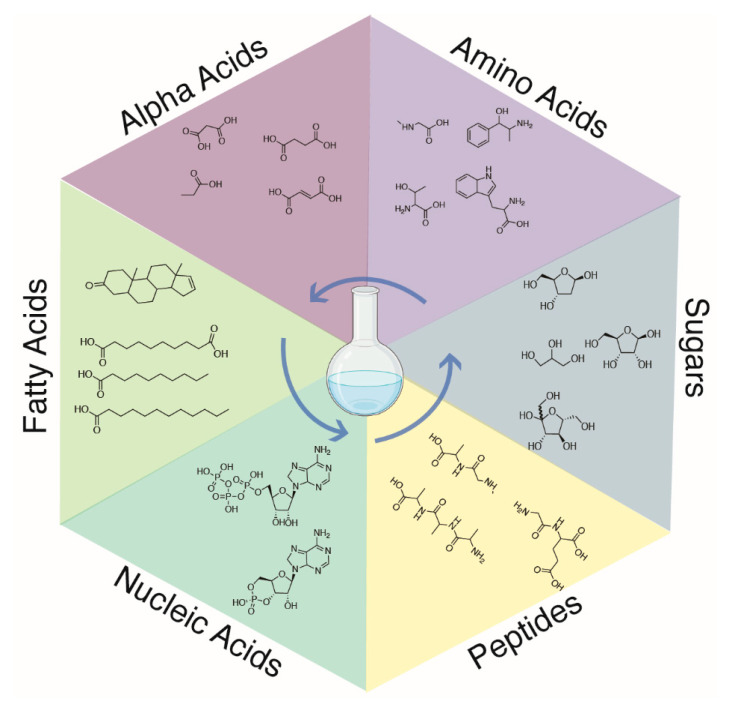
Overview of the molecules produced in the experiments. Multiple regassing cycles, as well as the addition of “sea water” supplemented with calcium phosphate and magnesium sulfate, led to the production of a diverse set of prebiotic molecules in a “one pot” synthesis.

**Figure 2 life-13-00265-f002:**
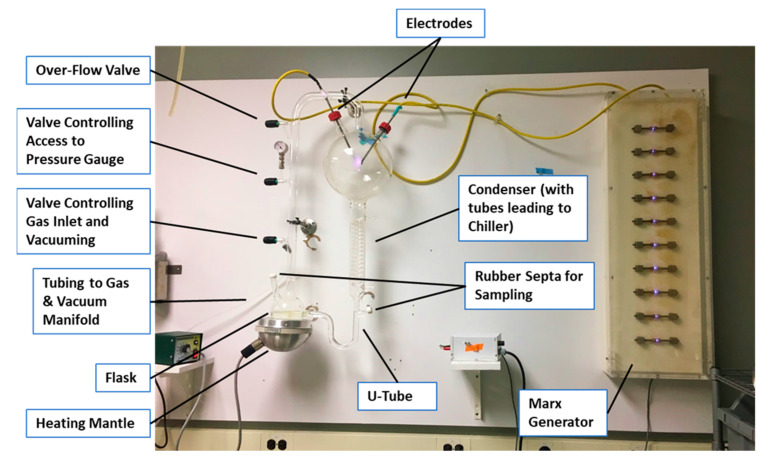
The modified Miller apparatus used in the experiments.

**Figure 3 life-13-00265-f003:**
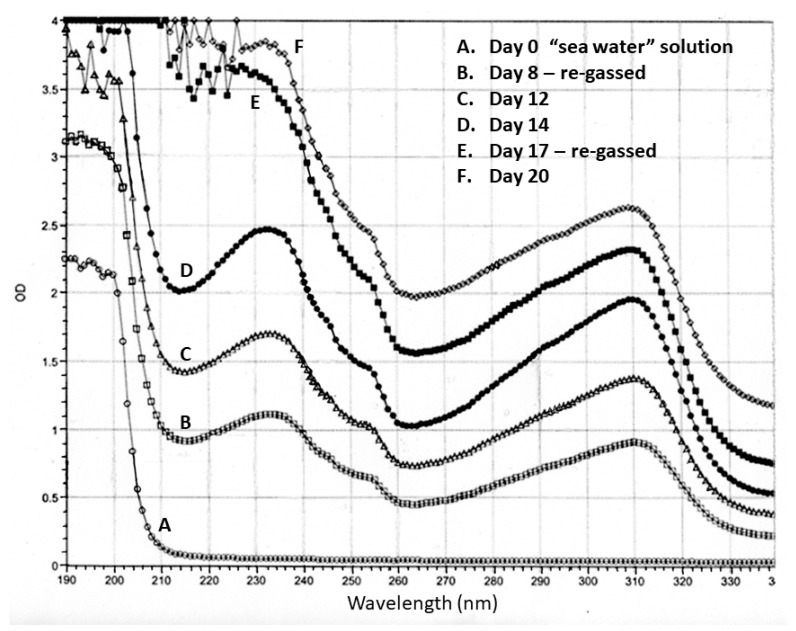
Ultraviolet spectroscopy of samples from the flask of the apparatus taken prior to turning the apparatus on (A) and at days 8 (B), 12 (C), 14 (D), 17 (E), and 20 (F). The apparatus was regassed on days 8 (C) and 17 (E) after the samples were taken. The original Miller experiment ended about day 8 (B) [3], and very few other long-term Miller-type experiments have been run since (reviewed in [4].

**Figure 4 life-13-00265-f004:**
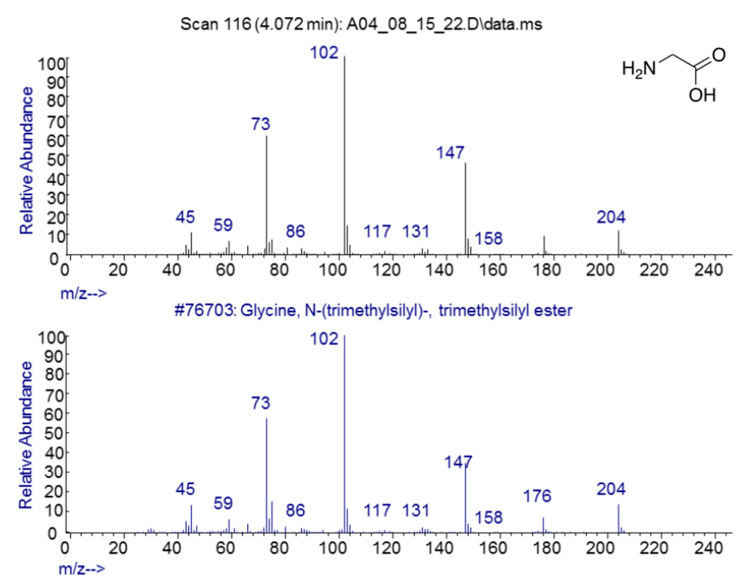
GC–MS spectra of MSTFA-pyridine prepared pure glycine (bottom) and glycine identified in sample from the apparatus after nine days of incubation.

**Figure 5 life-13-00265-f005:**
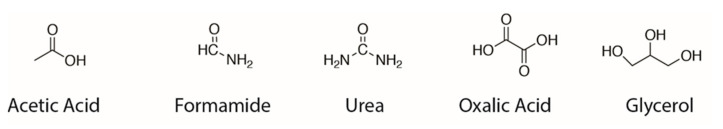
“Reactant compounds”, from which many more complex prebiotic molecules can be synthesized, were found in all of the experiments.

**Figure 6 life-13-00265-f006:**
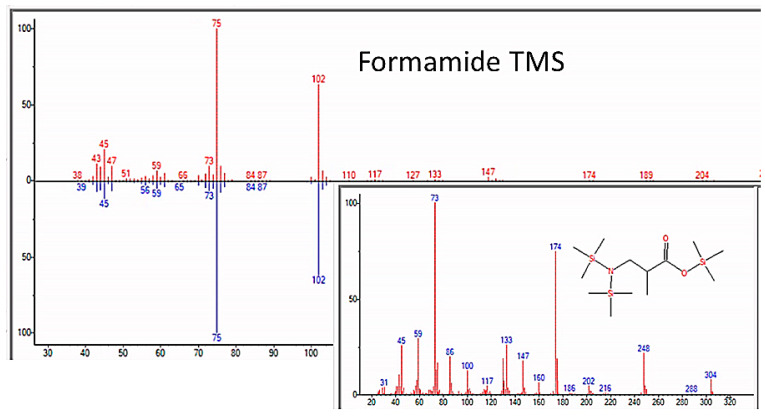
GC–MS identification of formamide from reaction mixture using the NIST database recognition software.

**Figure 7 life-13-00265-f007:**
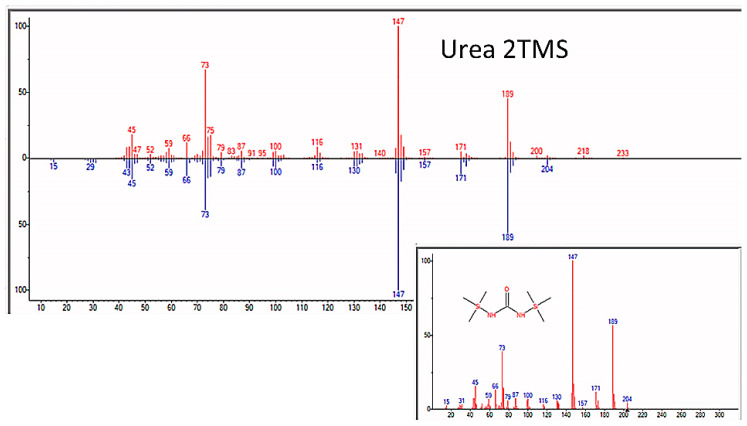
GC–MS identification of urea from reaction mixture using the NIST database recognition software.

**Figure 8 life-13-00265-f008:**
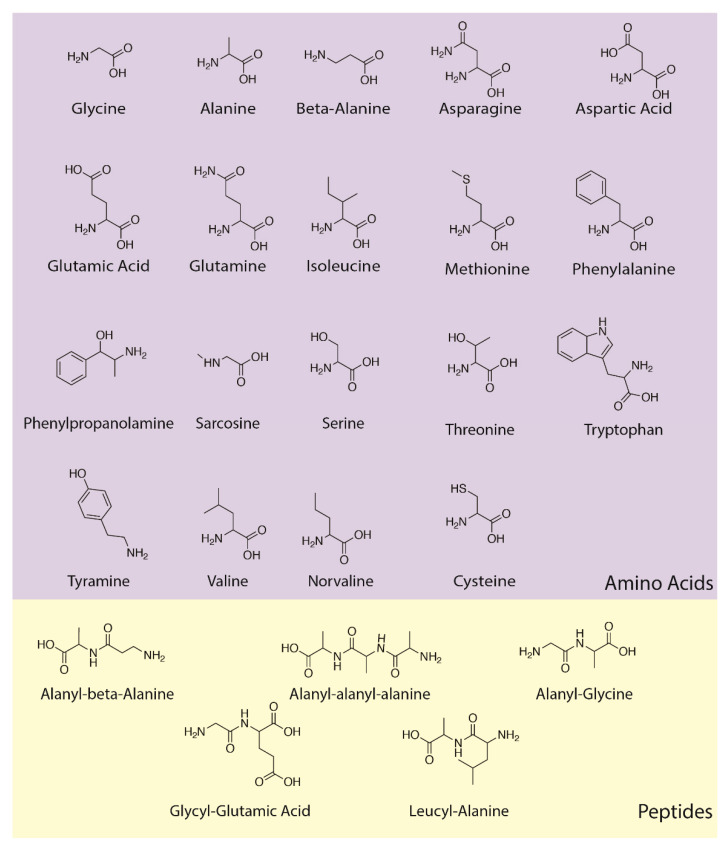
Overview of amino acids and peptides produced in this study. These compounds were undoubtedly racemic, and thus, the use of a non-chiral structural formalism here.

**Figure 9 life-13-00265-f009:**
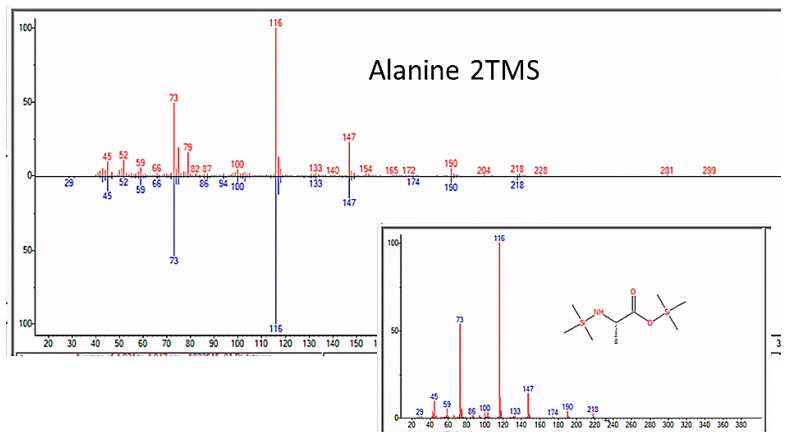
GC–MS identification of alanine using the NIST database recognition software.

**Figure 10 life-13-00265-f010:**
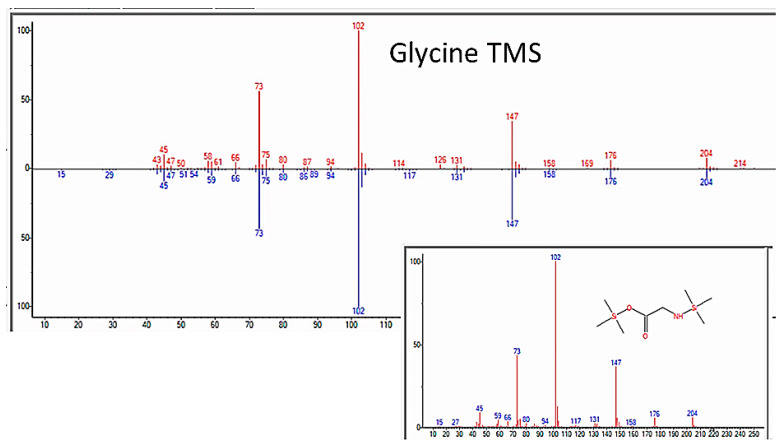
GC–MS identification of glycine from reaction mixture using the NIST database recognition software.

**Figure 11 life-13-00265-f011:**
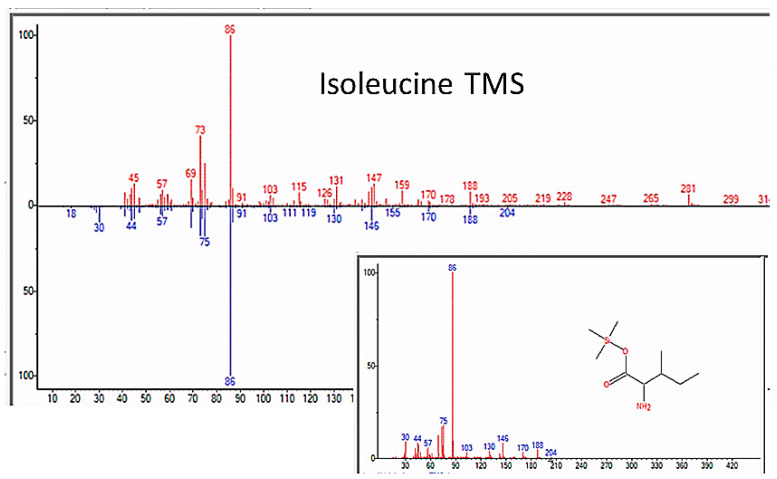
GC–MS identification of isoleucine from reaction mixture using the NIST database recognition software.

**Figure 12 life-13-00265-f012:**
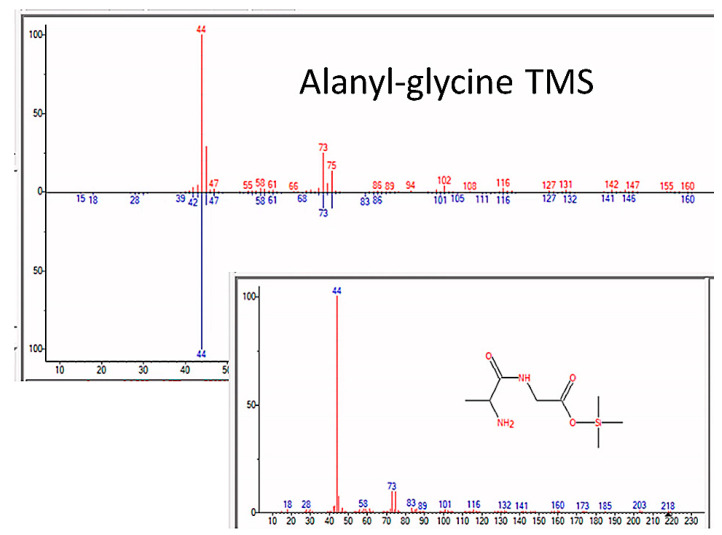
GC–MS identification of Alanyl–glycine from reaction mixture using the NIST database recognition software.

**Figure 13 life-13-00265-f013:**
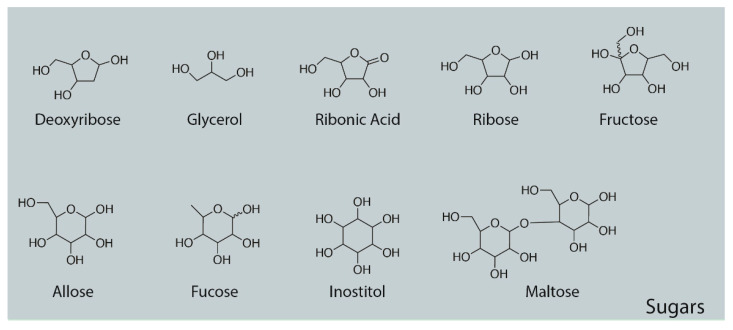
Sugars repeatedly identified by GC–MS in the experiments. Both D- and L- sugars were undoubtedly produced, and so was the use of a non-chiral structural formalism in these structures.

**Figure 14 life-13-00265-f014:**
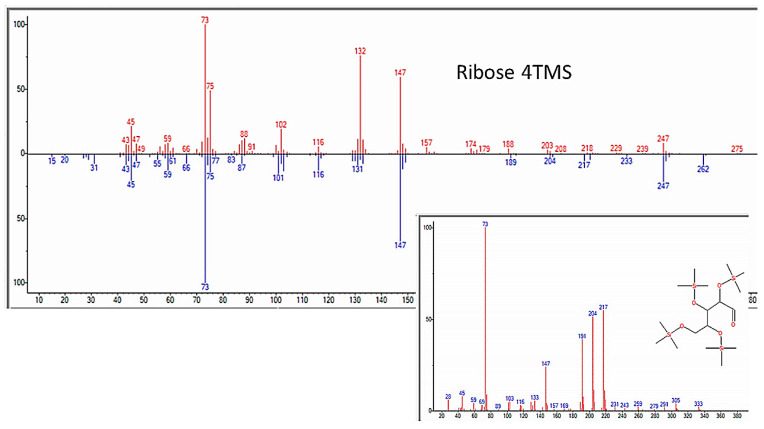
GC–MS identification of ribose from reaction mixture using the NIST database recognition software.

**Figure 15 life-13-00265-f015:**
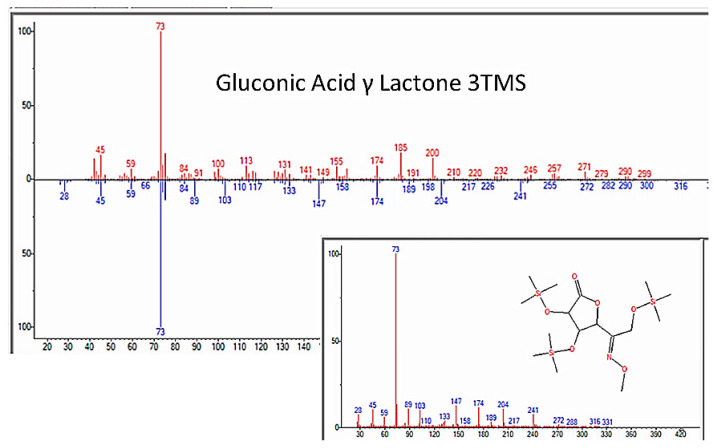
GC–MS identification of one of several glucose metabolites from reaction mixture using the NIST database recognition software.

**Figure 16 life-13-00265-f016:**
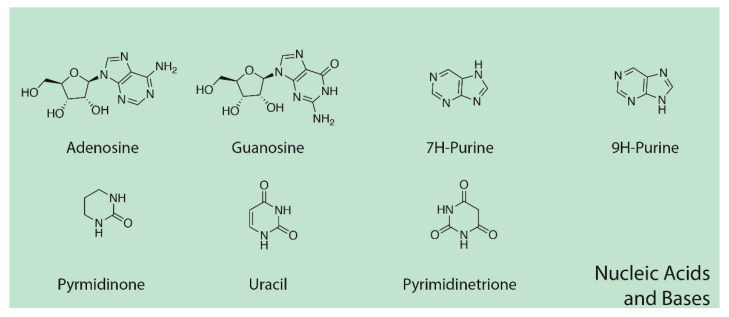
Nucleic acid bases identified by GC–MS in the experiments. The figures shown here illustrate D-adenosine and D-guanosine, but both D- and L- isomers of these molecules were presumably produced since there is no chiral selection mechanism involved in our experiments.

**Figure 17 life-13-00265-f017:**
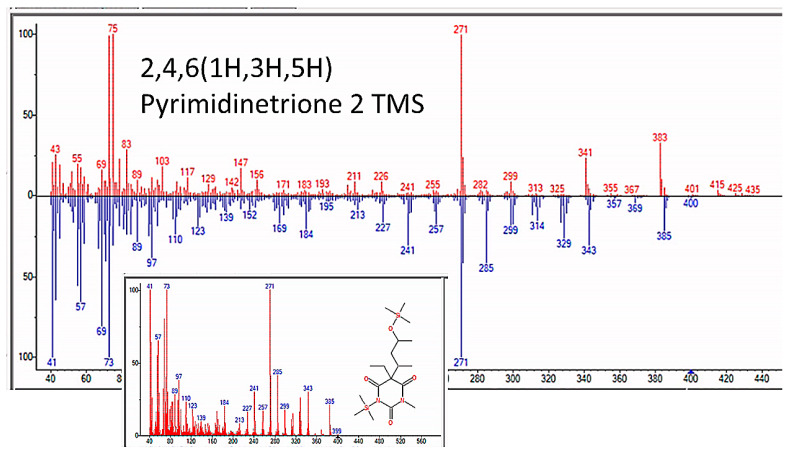
GC–MS identification of one of several pyrimidine metabolites from reaction mixture using the NIST database recognition software.

**Figure 18 life-13-00265-f018:**
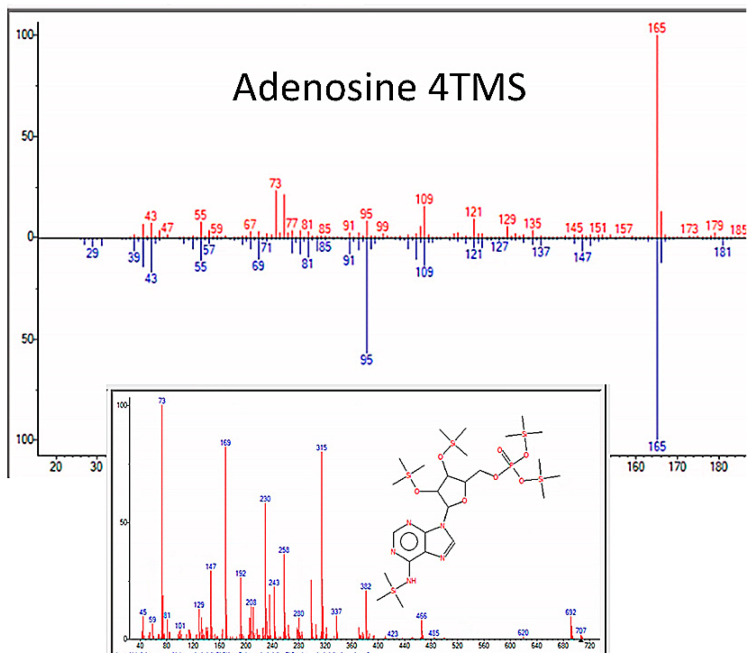
GC–MS identification of one of adenosine from reaction mixture using the NIST database recognition software.

**Figure 19 life-13-00265-f019:**
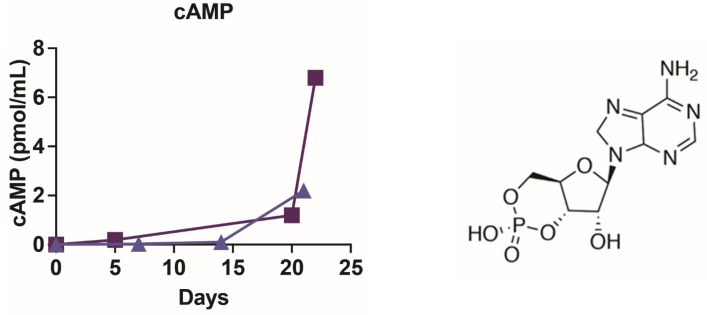
High sensitivity cyclic adenosine monophosphate ELISA results for two independent experiments, indicated by solid squares or triangles. Since this was an antibody assay, it presumably reacted only to the D-enantiomer of the compound, although there is every reason to believe the reaction mixture contained the L-enantiomer as well.

**Figure 20 life-13-00265-f020:**
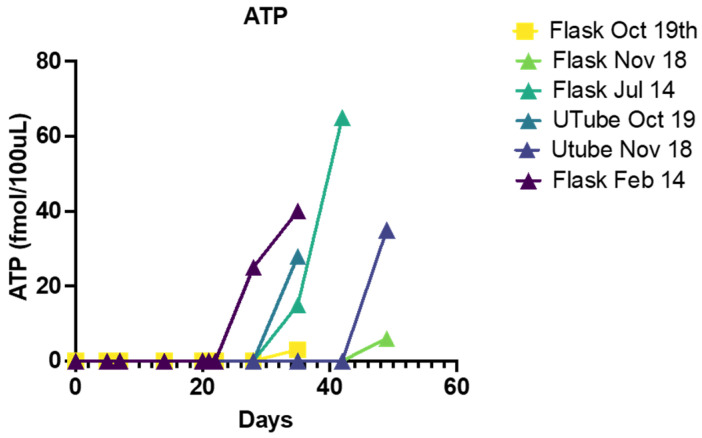
Adenosine triphosphate (ATP) readings from four independent experiments (two comparing values from the flask to the u-tube sources). No ATP was observed from any experiment until 28 days or thereafter, and all values were confirmed by multiple measurements (between 2 and 6) and accepted only if control materials (deionized, sterilized water, and the original “sea water”) sources tested repeatedly negative for ATP. Since this was an enzyme assay, it presumably reacted only to the D-enantiomer of the compound, although there is every reason to believe the reaction mixture contained the L-enantiomer as well.

**Figure 21 life-13-00265-f021:**
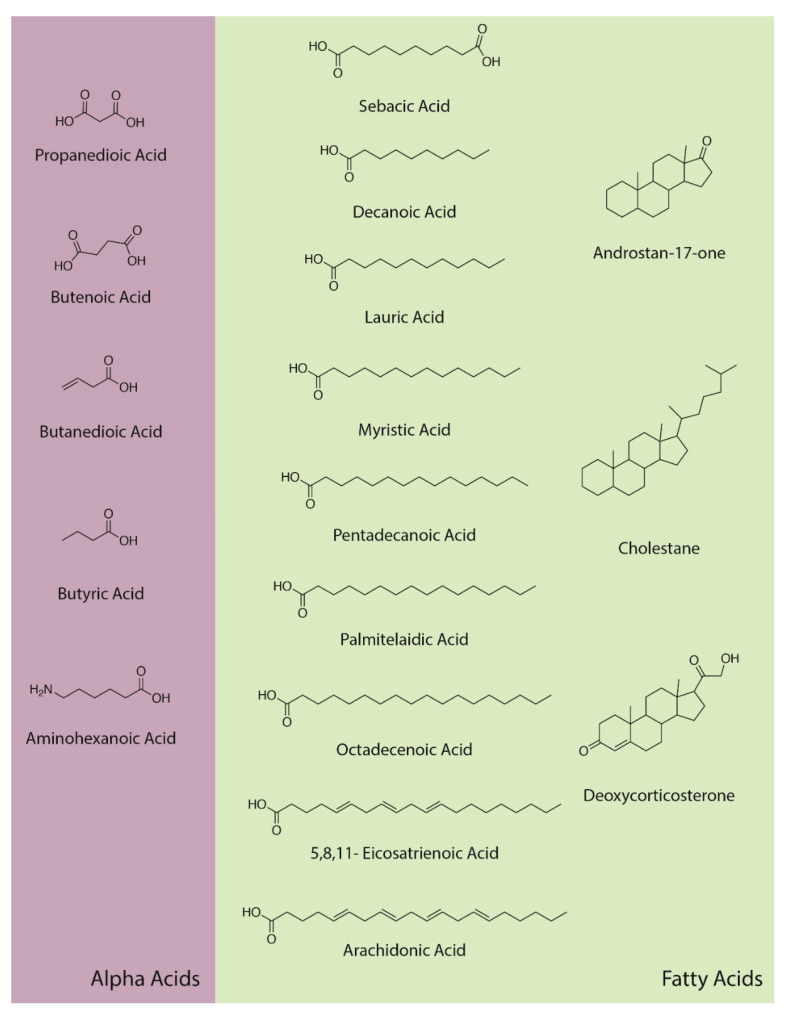
Alpha acids, fatty acids, and steroidal compounds identified by GC–MS in the experiments.

**Figure 22 life-13-00265-f022:**
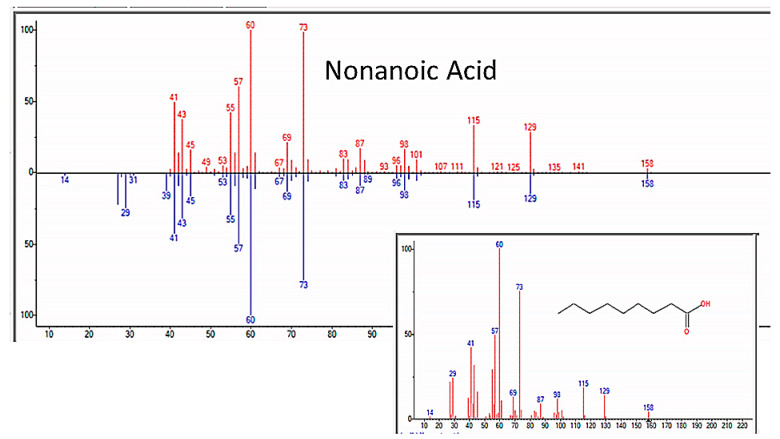
GC–MS identification of nonanoic acid from reaction mixture using the NIST database recognition software.

**Figure 23 life-13-00265-f023:**
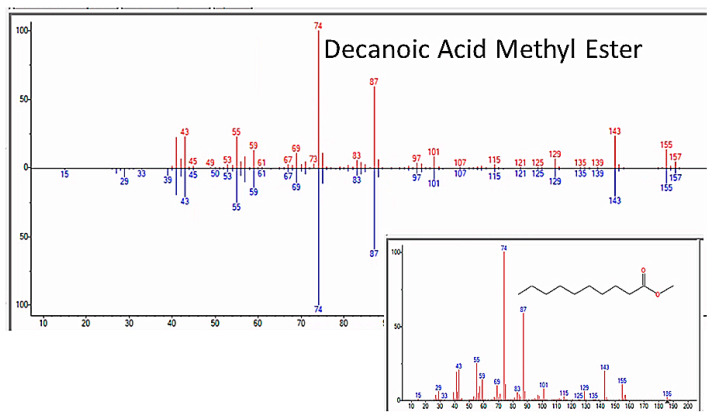
GC–MS identification of decanoic acid from reaction mixture using the NIST database recognition software.

**Figure 24 life-13-00265-f024:**
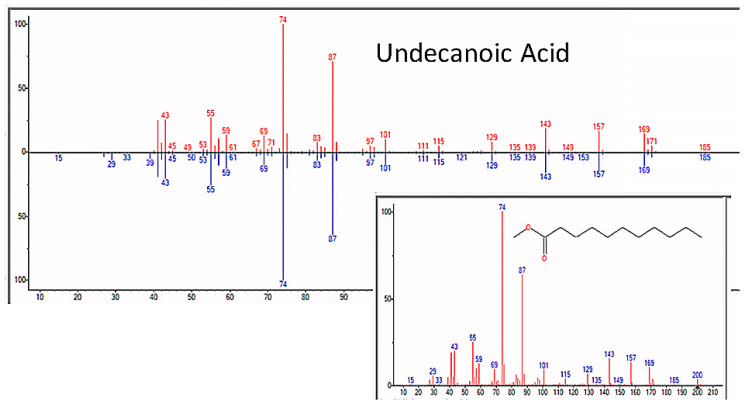
GC–MS identification of undecanoic acid from reaction mixture using the NIST database recognition software.

**Figure 25 life-13-00265-f025:**
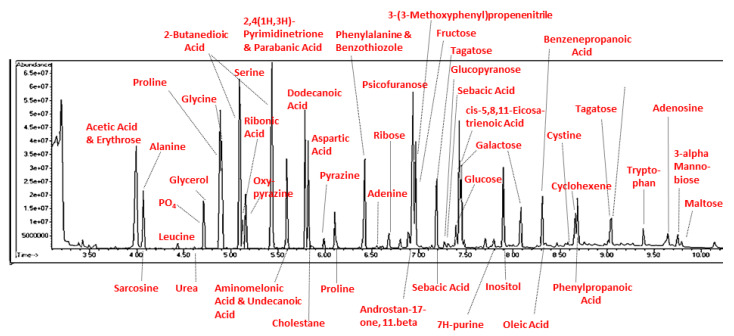
Chromatogram of end of week three sample (u-tube) labeled with the NIST-identified products.

**Figure 26 life-13-00265-f026:**
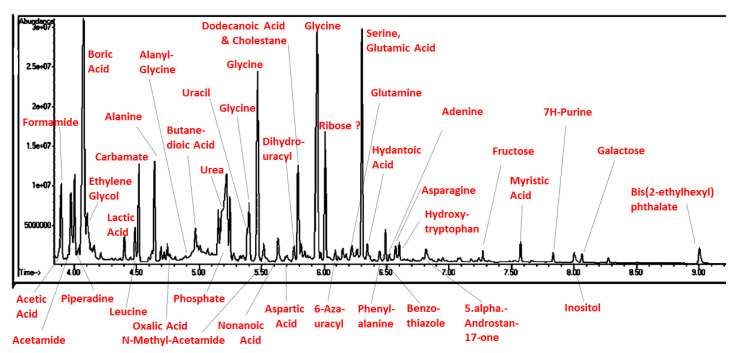
Chromatogram of end of week three sample (u-tube) labeled with the NIST-identified products.

**Figure 27 life-13-00265-f027:**
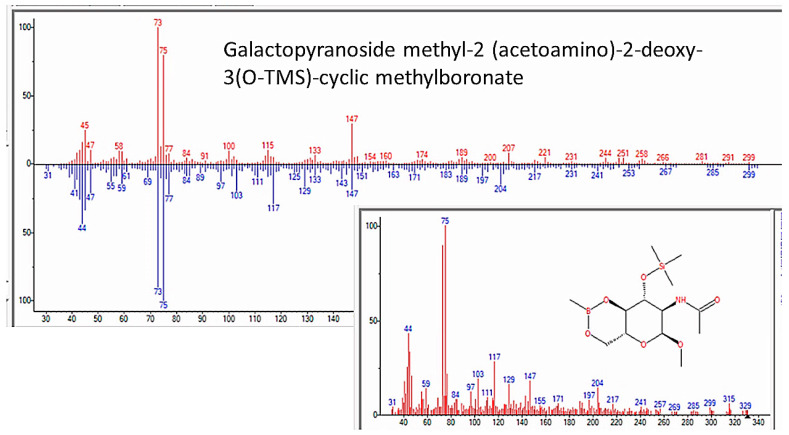
GC–MS identification of 6-azauracil from reaction mixture using the NIST database recognition software.

**Figure 28 life-13-00265-f028:**
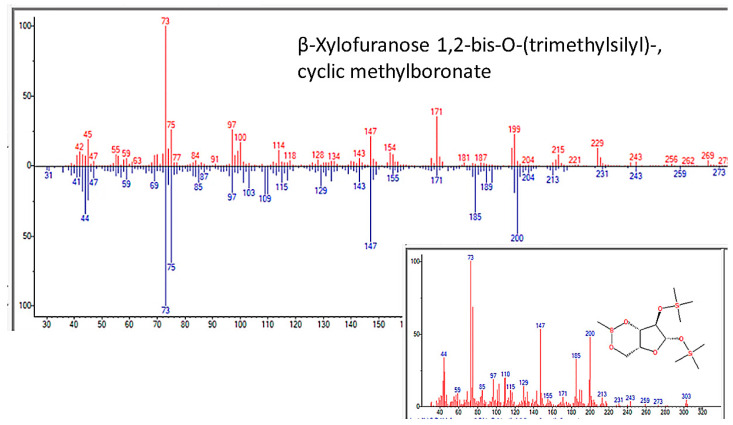
GC–MS identification of 6-azauracil from reaction mixture using the NIST database recognition software.

**Figure 29 life-13-00265-f029:**
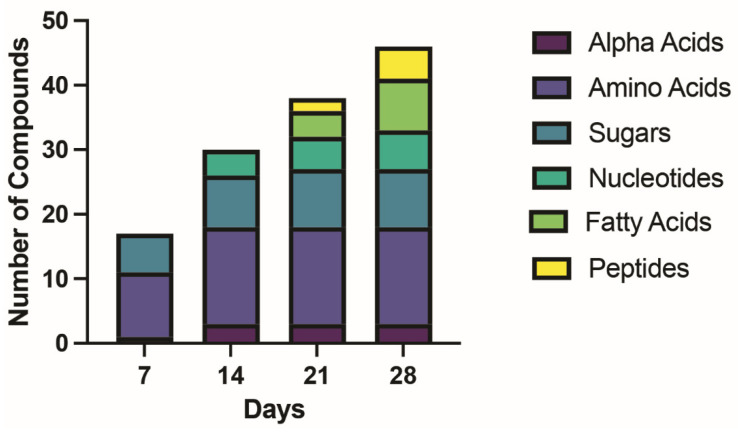
Time course of appearance of different classes of compounds. The *y* axis is the number of compounds where each color is type.

**Table 1 life-13-00265-t001:** “Reactant compounds”, from which many more complex prebiotic molecules can be synthesized, were found in all of the experiments. “RT” is retention time in minutes (min). “Mol. Wt.” is the molecular weight of the identified compound in grams per mole. “QOM” is the quality of match out of 100, according to the NIST database matching software.

Reactant Compounds	RT (min)	Hit Name	Mol. Wt.	QOM
Acetic Acid	3.851	Acetic acid, [(trimethylsilyl)oxy]-, trimethylsilyl ester	220.41	80
Formamide	3.982	Formamide, (trimethylsilyl)-	189.4	87
Glycerol	4.751	Glycerol, tris(trimethylsilyl) ether	308.64	90
Oxalic Acid	4.783	Oxalic acid, di(trimethylsilyl)	234.40	87
Urea	4.327	Urea, (trimethylsilyl)-	426.68	96

**Table 2 life-13-00265-t002:** Summary of amino acids identified by GC–MS repeatedly in at least two independent experiments. “QOM” is quality of match out of 100 as determined by the NIST database. “RT” is retention time in minutes (min). “Mol. Wt.” is the molecular weight of the identified compound in grams per mole. Note that the QOMs for Cystathione and proline are much lower than for the other compounds identified in the table, and the reliability of the identification is correspondingly lower. However, the presence of both cysteine and proline with higher QOM suggests that these identifications are plausible.

Amino Acids	RT (min)	Hit Name	Mol. Wt.	QOM
Alanine	4.080	Alanine, N-(trimethylsilyl)-, trimethylsilyl ester	233.45	93
Asparagine	6.525	Asparagine, N,N2-bis(trimethylsilyl)-, trimethylsilyl ester	348.66	99
Aspartic Acid	5.821	Aspartic acid, N-(trimethylsilyl)-, bis(trimethylsilyl) ester	349.64	98
Beta-alanine	5.867	Beta-alanine, N-(trimethylsilyl)-, bis(trimethylsilyl) ester	305.64	75
Cystathionine	9.777	Cystathionine, N-(trimethylsilyl)-, bis(trimethylsilyl) ester	366.63	51
Cysteine	8.622	Cysteine, N,N’-bis(trimethylsilyl)-, bis(trimethylsilyl) ester	337.70	76
Glutamic Acid	6.313	Glutamic acid, N-(trimethylsilyl)-, bis(trimethylsilyl) ester	363.67	98
Glutamine	6.924	Glutamine, tris(trimethylsilyl)	362.69	64
Glycine	4.929	Glycine, N,N-bis(trimethylsilyl)-, trimethylsilyl ester	291.61	86
Hydroxytryptophan	4.581	5-Hydroxytryptophan, tetramethylsilylester	508.95	87
Isoleucine	8.588	Isoleucine, N-(trimethylsilyl)-, trimethylsilyl ester	275.53	81
Leucine	4.42	Leucine, trimethylsilyl ester	203.35	95
Methionine	10.311	Methionine-(trimethylsilyl)	221.39	72
Oxyproline	5.965	Proline, 5-oxo-1-(trimethylsilyl)-, trimethylsilyl ester	275.49	74
Phenylalanine	6.415	Phenylalanine, N,O-Bis-(trimethylsilyl)	309.55	91
Phenylpropanolamine	4.080	Phenylpropanolamine, bis(trimethylsilyl)	295.57	80
Proline	10.651	Proline, trimethylsilyl)-, trimethylsilyl ester	259.49	57
Sarcosine	4.165	Sarcosine, Bis(trimethylsilyl)	233.45	81
Serine	5.201	Serine, N,O-bis(trimethylsilyl)-, trimethylsilyl ester	321.63	87
Threonine	5.320	Threonine, N,O,O-Tris(trimethylsilyl)-	335.66	91
Tryptophan	9.420	Tryptophan, bis(trimethylsilyl)-	348.6	89
Tyramine	5.707	Tyramine, tri(trimethylsilyl)-	353.72	90
Valine	4.581	Valine, N-(trimethylsilyl)-, trimethylsilyl ester	261.51	90
Norvaline	4.581	Norvaline, N-(trimethylsilyl)-, trimethylsilyl ester	261.51	83

**Table 3 life-13-00265-t003:** Summary of peptides identified by GC–MS in independent experiments. “RT” is retention time in minutes (min). “Mol. Wt.” is the molecular weight of the identified compound in grams per mole. “QOM” is quality of match out of 100 as determined by the NIST database.

Peptides	RT	Hit Name	Mol. Wt.	QOM
Alanyl-beta-alanine	5.201	Alanyl-beta-alanine, N,O-bis(trimethylsilyl)-, trimethylsilyl ester	232.35	87
Alanyl–alanyl–alanine	9.768	Alanyl–alanyl–alanine methyl ester	245.28	59
Alanyl–glycine	6.0242	Alanyl–glycine, bis(trimethylsilyl) ester	218.33	87
Glycyl-glutamic acid	17.286	Glycyl-glutamic acid, bis(trimethylsilyl) ester	348.54	86
Leucyl–alanine	8.588	Leucyl–alanine, bis(trimethylsilyl) ester	376.69	81

**Table 4 life-13-00265-t004:** Summary of sugars repeatedly identified by GC–MS in the experiments. “RT” is retention time in minutes (min). “Mol. Wt.” is the molecular weight of the identified compound in grams per mole. “QOM” is the quality of match out of 100 according to the NIST database matching software.

Monosaccharides	RT (min)	Hit Name	Mol. Wt.	QOM
Allose	7.4	Allose, pentakis(trimethylsilyl) ether, methyloxime (anti)	570.10	90
Deoxyribose	4.072	2-Deoxy-ribose, tris(trimethylsilyl) ether	437.87	53
Fructose	7.221	Fructose, pentakis(trimethylsilyl) ether, methyloxime (anti)	570.10	97
Fucose	7.552	Fucose, tetrakis(trimethylsilyl) ether	481.90	91
Galactopyranose	7.552	Galactopyranose, pentakis(trimethylsilyl) ether (isomer 2)	541.06	90
Galactose	7.4	Galactose, 2,3,4,5,6-pentakis-O-(trimethylsilyl)-, o-methyloxyme, (1E)-	570.10	91
Glucose	7.4	Glucose, 2,3,4,5,6-pentakis-O-(trimethylsilyl)-, o-methyloxyme, (1Z)-	628.30	87
Gluconic Acid	8.792	Gluconic acid, 2,3,4,6-tetrakis-O-(trimethylsilyl)-, .delta.-lactone	466.86	77
Inositol	7.994	Inositol, 1,2,3,4,5,6-hexakis-O-(trimethylsilyl)-, cis-	613.24	90
Levoglucosan	6.67	Levoglucosan, tris(trimethylsilyl)-	378.68	87
Lyxose	7.221	Lyxose, tetrakis(trimethylsilyl) ether, trimethylsilyloxime (isomer 1)	526.05	90
Mannose	7.4	Mannose, 2,3,4,5,6-pentakis-O-(trimethylsilyl)-, o-methyloxyme, (1Z)-	570.10	91
Myo-Inositol	8.002	Myo-Inositol, 1,2,3,4,5,6-hexakis-O-(trimethylsilyl)-	613.24	99
Ribonic Acid	5.091	Ribonic acid, 5-deoxy-2,3-bis-O-(trimethylsilyl)-, .gamma.-lactone	276.48	89
Ribose	6.355	Ribose, 2,3,4,5-tetrakis-O-(trimethylsilyl)-	438.9	87
Ribo-hexos-3-ulose	4.700	Ribo-hexos-3-ulose, 2,4,5,6-tetrakis-O-(trimethylsilyl)-, bis(O-methyloxime)	525.0	73
Ribono-1,4-lactone	9.352	Ribono-1,4-lactone, tris(trimethylsilyl) ether	364.65	63
Sorbitol	7.434	Sorbitol, hexakis(trimethylsilyl) ether	615.26	89
Sorbose	7.221	Sorbose, pentakis(trimethylsilyl) ether, trimethylsilyloxime (isomer 1)	628.26	90
Tagatose	7.221	Tagatose, pentakis(trimethylsilyl) ether, trimethylsilyloxime	628.26	91
Talose	7.4	Talose, pentakis(trimethylsilyl) ether, methyloxime (syn)	570.10	91

**Table 5 life-13-00265-t005:** Summary of disaccharides identified in reaction mixtures using the NIST database recognition software. “RT” is retention time in minutes (min). “Mol. Wt.” is the molecular weight of the identified compound in grams per mole. “QOM” is the quality of match out of 100 according to the NIST database matching software.

Disaccharides	RT (min)	Hit Name	Mol. Wt.	QOM
Maltose	9.836	Maltose, octakis(trimethylsilyl) ether, methyloxime (isomer 1)	948.78	93
Trehalose	9.836	Trehalose, octakis(trimethylsilyl) ether	919.75	64

**Table 6 life-13-00265-t006:** Summary of nucleic acid bases and precursors repeatedly identified by GC–MS in the experiments. “RT” is retention time in minutes (min). “Mol. Wt.” is the molecular weight of the identified compound in grams per mole. “QOM” is the quality of match out of 100 according to the NIST database matching software.

Nucleic Acid Precursors	RT (min)	Hit Name	Mol. Wt.	QOM
Adenine	6.551	Adenine, N,7-bis(trimethylsilyl)-	279.49	60
Adenosine	8.461	Adenosine, N-(4-hydroxy-3-methyl-2-butenyl)-, (E)-	351.15	80
Adenosine	9.598	Adenosine-tetrakis(trimethylsilyl)-	555.97	93
Guanine	7.824	Thioguanine	167.19	59
Guanosine	9.972	Guanosine,N-Methyl penta(trimethylsilyl)-	644.10	56
7H-Purine	7.799	7-(Trimethylsilyl)-2,6-bis[(trimethylsilyl)oxy]-7H-purine	368.65	99
7H-Purine	7.714	7-(Trimethylsilyl)-2,6-bis[(trimethylsilyl)oxy]-7H-purine	368.65	91
9H-Purine	6.271	9H-Purine, 9-(trimethylsilyl)-2,6-bis[(trimethylsilyl)oxy]-	368.65	78
Pyrimidinetrione	7.484	2,4,6(1H,3H,5H)-Pyrimidinetrione, 5-[2-(methoxyimino)-3-[(trimethylsilyl)]-	399.6	93
Pyrimidine	7.077	Pyrimidine, 2,4,6-tris[(trimethylsilyl)oxy]-	344.63	76
Pyrimidine	7.391	1,2,4-Triazolo[1,5-a]pyrimidine, 5,7-dimethyl-2-phenyl-	224.26	64
Pyrmidinone	5.345	5-Methyldihydro-2,4(1H,3H)-pyrimidinedione diTMS	272.49	70
Pyrmidinone	8.962	2(1H)-Pyrimidinone, 5-(4-methylphenoxy)-4-(4-nitrophenyl)-6-phenyl-	399.40	64
Uracil	5.779	Dihydro-uracil-di(trimethylsilyl)-	258.46	69
Uracil	5.209	6-Azauracil, bis(tert-butyldimethylsilyl) deriv.	341.6	72

**Table 7 life-13-00265-t007:** Summary of steroidal and fatty acid compounds identified by GC–MS in the experiments. “RT” is retention time in minutes (min). “Mol. Wt.” is the molecular weight of the identified compound in grams per mole. “QOM” is quality of match out of 100 as determined by the NIST database.

Steroids	RT	Hit Name	Mol. Wt.	QOM
Androstan-17-one	16.462	Androstan-17-one, 3-[(trimethylsilyl)oxy]-, (3.alpha.,5.alpha.)-	362.6	97
Androstan-17-one	6.848	5.alpha.-Androstan-17-one, 11.beta.-hydroxy-3.alpha.-(trimethylsiloxy)-	378.6	98
Androstan-17-one	8.461	5.beta.-Androstan-17-one, 11.beta.-hydroxy-3.alpha.-(trimethylsiloxy)-	378.6	95
Cholestane	5.88	Cholestane, 2,3-epoxy-, (2.alpha.,3.alpha.,5.alpha.)-	386.7	92
Deoxycorticosterone	8.919	4-Pregnen-21-ol-3,20-dione glucoside	492.6	95
ALPHA AND FATTY ACIDS	RT	Hit Name	Mol. Wt.	QOM
Propanedioic Acid, Malonic Acid (C3)	9.53	Propanedioic acid, (1H-indole-3-ylmethylene)-, diethyl ester	287.31	93
Butenoic Acid, Succinic Acid (C4)	6.186	2-Butenoic acid, 3-methyl-2-[(trimethylsilyl)oxy]-, trimethylsilyl ester	260.48	76
Butanedioic Acid, Fumeric Acid (C4)	5.048	Butanedioic acid, bis(trimethylsilyl) ester	262.45	97
Butyric Acid (C4)	6.024	2,3,4-Trihydroxybutyric acid tetrakis(trimethylsilyl) deriv., (, (R*,R*)-)	424.8	87
Aminohexanoic Acid (C6)	4.581	N,O,O’-Tris-(trimethylsilyl)-6-hydroxy-2-aminohexanoic acid	363.71	90
Sebacic Acid (C10)	7.349	Sebacic acid, bis(trimethylsilyl) ester	346.61	87
Decanoic Acid (C10)	9.318	Decanoic acid, 2-[(trimethylsilyl)oxy]-1-[[(trimethylsilyl)oxy]methyl]ethyl ester	390.7	62
Lauric Acid (C12)	12.985	12-Methylaminolauric acid	229.36	59
Myristic Acid (C14)	7.061	Myristic acid, trimethylsilyl ester	300.6	89
Pentadecanoic Acid (C15)	7.394	Pentadecanoic acid, trimethylsilyl ester	314.6	80
Palmitelaidic Acid (C16)	7.773	Palmitelaidic acid, trimethylsilyl ester	326.6	62
Octadecenoic Acid, Stearic Acid (C18)	7.484	9-Trimethylsilyloxy-12-octadecenoic acid, methyl ester	384.7	80
Arachidonic Acid (C20)	7.604	Arachidonic acid, trimethylsilyl ester	376.6	91
5,8,11-Eicosatrienoic acid (C20)	7.255	cis-5,8,11-Eicosatrienoic acid, trimethylsilyl ester	378.7	91

**Table 8 life-13-00265-t008:** Time course of appearance of classes of compounds. All compounds listed here were observed at least twice (and usually more often) in independent experiments and had high–quality identification scores from GC–MS or other assays. Exceptions are noted with a question mark following the compound name, indicating that the identification of the compound was not repeatable or had a low identification quality. ”Times” refers to how many regassings had occurred (usually done every seven or eight days).

	Day 0	1 Time	2 Times	3 Times	4 Times	5 Times
Phosphates	0	25 ppm	50 ppm	75 ppm	100 ppm	150 ppm
Sugars		Fructose, Galactose, Glucose, Mannose, Sorbose, Tagatose	Xylose, Fucose, Maltose Trehalose?	Ribose, Deoxyribose?		
Amino Acids		Ala, Asn, Asp, Gln, Glu, Gly, Ile, Leu, Ser, Trp	Phenylpropanolamine (amphetamine), Cys, Met, Val, Nor-Val		Ala-Gly Gly-Glu	Ala-Ala Ala-Ala-Ala Leu-Ala
Nucleic Acids			Adenosine, Guanosine? 7H-Purine, 9H-Purine, Pyrimidinones	cAMP, cGMP?	Uracil	ATP
Fatty Acids			Aminohexanoic acid (C6), Butanoic Acid (C4), (Succinic Acid), Butyric Acid (C4)	Sebacic acid (C10), Decanoic Acid (C10), Arachidonic Acid (C20), Eicosatrianoic Acid (C20)	Lauric Acid (C14), Myristic Acid (C14), Octadecenoic Acid (C18), Steroids	Hexadecanoic Acid (C16) Palmitalaidic Acid (C16)

## Data Availability

Original data are available by application to the corresponding author.

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
