# Peer review of "“Sea Water” Supplemented with Calcium Phosphate and Magnesium Sulfate in a Long-Term Miller-Type Experiment Yields Sugars, Nucleic Acids Bases, Nucleosides, Lipids, Amino Acids, and Oligopeptides"

_life, 2023, doi:10.3390/life13020265_

Round 1

Reviewer 1 Report (Previous Reviewer 1)

The authors improved the manuscript and for this, I agree to accept it as is for publication.  Please do the corresponding correction in "alanlyl-glycine" in line 385.

Author Response

REVIEWER 1

Open Review

English language and style

( ) English very difficult to understand/incomprehensible
( ) Extensive editing of English language and style required
( ) Moderate English changes required
(x) English language and style are fine/minor spell check required
( ) I don't feel qualified to judge about the English language and style

Yes

Can be improved

Must be improved

Not applicable

Does the introduction provide sufficient background and include all relevant references?

(x)

( )

( )

( )

Are all the cited references relevant to the research?

(x)

( )

( )

( )

Is the research design appropriate?

( )

(x)

( )

( )

Are the methods adequately described?

(x)

( )

( )

( )

Are the results clearly presented?

(x)

( )

( )

( )

Are the conclusions supported by the results?

(x)

( )

( )

( )

Comments and Suggestions for Authors

The authors improved the manuscript and for this, I agree to accept it as is for publication.  Please do the corresponding correction in "alanlyl-glycine" in line 385.

LINE 385: FIXED!

Submission Date

20 December 2022

Date of this review

30 Dec 2022 04:00:45

Reviewer 2 Report (New Reviewer)

Title: “Sea Water” Supplemented with Calcium Phosphate and Magnesium Sulfate in a Long-Term Miller-Type Experiment Yields Sugars, Nucleic Acids, Lipids, Amino Acids and Peptides

Dear Authors, dear Editor

Overview and general recommendation:

The article is a novel approach in the field of prebiotic chemistry studies. For years, this area of study has focused on the synthesis of molecules relevant to living beings today. However, as explained in the article, such syntheses are clean and in no way represent the natural complexity in which the synthesis of organic molecules should have occurred on Earth.

The developed methodology, which includes the incorporation of multiple factors in a single device, is very promising in the field of prebiotic chemistry. The results presented are interesting and further experiments should be explored, including for example, other "primitive seas" compositions, and other minerals.

This paper is very relevant in the field, and it shows many and important data. I just want to consider the clarification of some points before publication.

Major comments:

1.      Tittle. I consider that a better title, that describes your findings, will be « “Sea Water” Supplemented with Calcium Phosphate and Magnesium Sulfate in a Long-Term Miller-Type Experiment Yields Sugars, Nucleic Acids Bases, Nucleosides, Lipids, Amino Acids and Oligopeptides.»

2.      All the spectra need to be improved, as they are hard to read, please change colours and/or size. I am not sure that all the spectra must be in the main text. Do you consider pertinent to include an appendix with all this information?

3.      Please, check along the text the numbering of Figures.

4.      Tables need to be in the same format (Font type and size) as the main text, please change it. Probably, the inclusion of the mass of the detected molecules will enhance the information of the summary tables.

5.      Explain how you select the quality of match (QOM) to discriminate the possible molecules formed.

6.      I’m curious, do you have data about the amounts of each formed compound in each sample?

7.      As the results are quite important, I suggest including, if possible, the amounts of the more abundant molecules detected.

Minor comments

Along the text I made some observations, please check it all.

Author Response

REVIEWER 2

Open Review

English language and style

( ) English very difficult to understand/incomprehensible
( ) Extensive editing of English language and style required
( ) Moderate English changes required
(x) English language and style are fine/minor spell check required
( ) I don't feel qualified to judge about the English language and style

Yes

Can be improved

Must be improved

Not applicable

Does the introduction provide sufficient background and include all relevant references?

(x)

( )

( )

( )

Are all the cited references relevant to the research?

(x)

( )

( )

( )

Is the research design appropriate?

(x)

( )

( )

( )

Are the methods adequately described?

(x)

( )

( )

( )

Are the results clearly presented?

( )

( )

(x)

( )

Are the conclusions supported by the results?

(x)

( )

( )

( )

Comments and Suggestions for Authors

Title: “Sea Water” Supplemented with Calcium Phosphate and Magnesium Sulfate in a Long-Term Miller-Type Experiment Yields Sugars, Nucleic Acids, Lipids, Amino Acids and Peptides

Dear Authors, dear Editor

Overview and general recommendation:

The article is a novel approach in the field of prebiotic chemistry studies. For years, this area of study has focused on the synthesis of molecules relevant to living beings today. However, as explained in the article, such syntheses are clean and in no way represent the natural complexity in which the synthesis of organic molecules should have occurred on Earth.

The developed methodology, which includes the incorporation of multiple factors in a single device, is very promising in the field of prebiotic chemistry. The results presented are interesting and further experiments should be explored, including for example, other "primitive seas" compositions, and other minerals.

This paper is very relevant in the field, and it shows many and important data. I just want to consider the clarification of some points before publication.

Major comments:

  1. I consider that a better title, that describes your findings, will be « “Sea Water” Supplemented with Calcium Phosphate and Magnesium Sulfate in a Long-Term Miller-Type Experiment Yields Sugars, Nucleic Acids Bases, Nucleosides, Lipids, Amino Acids and Oligopeptides.»

MODIFIED AS SUGGESTED.

  1. All the spectra need to be improved, as they are hard to read, please change colours and/or size. I am not sure that all the spectra must be in the main text. Do you consider pertinent to include an appendix with all this information?

THE PROBLEM IS NOT WITH THE QUALITY OF THE SPECTRA, WHICH ARE HIGH QUALITY TIFF FILES, BUT WITH THE CONVERSION TO PDF, OVER WHICH WE HAVE NO CONTROL THERE IS NO WAY TO CHANGE THE COLORS: THAT IS HOW THE MASS SPEC PROGRAM DELIVERS THE DATA. AS FOR SIZE, OUR EXPERIENCE IS THAT THE IMAGES WILL BE MADE INTO POP-OUTS IN THE PUBLISHED VERSION, WHICH MEANS THAT THE CAN BE ENLARGED TO THEIR ORIGINAL SIZE AND QUALITY.  WE WILL WORK WITH EDITORS TO ENSURE BETTER QUALITY!

  1. Please, check along the text the numbering of Figures.

DONE! THANK YOU FOR THE HEADS-UP ABOUT NUMBERING PROBLEM! (THEY ALL HAD TO RE-NUMBERED ANYWAY DUE TO SHIFTING MANY TO APPENDICES….HOPEFULLY NO NEW ERRORS WERE INTRODUCED!) 

  1. Tables need to be in the same format (Font type and size) as the main text, please change it.

DONE

Probably, the inclusion of the mass of the detected molecules will enhance the information of the summary tables.

DONE

  1. Explain how you select the quality of match (QOM) to discriminate the possible molecules formed.

DESCRIPTION OF QOM CRITERIA ADDED TO METHODS

  1. I’m curious, do you have data about the amounts of each formed compound in each sample?

WE DO NOT HAVE ACCURATE DATA ON AMOUNTS OF EACH COMPOUND FORMED IN EACH SAMPLE, THOUGH SOME SENSE OF RELATIVE CONCENTRATIONS CAN BE GAINED FROM THE PEAK HEIGHTS IN THE ORIGINAL CHROMATOGRAMS. PART OF THE DIFFICULTY IS THAT COMPOUND SPECTRA OVERLAP CREATING SERIOUS DIFFICULTIES IN DETERMINING HOW MUCH OF EACH OBSERVED PEAK IS DUE TO THE CONTRIBUTION OF ANY PARTICULAR COMPOUND. ADDITIONALLY, IN THE ABSENCE OF TITRATION SPECTRA FOR EACH  DERIVITIZED PURE COMPOUND WITH WHICH TO COMPARE THE EXPERIMENTAL RESULTS – WHICH IS PRESENTLY BEYOND OUR MEANS – WE ARE LOATH TO SAY ANYTHING SPECIFIC ABOUT CONCENTRATIONS. WE HAVE, HOWEVER, ADDED THIS POINT AS A LIMITATION IN OUR CONCLUDING SECTION.

  1. As the results are quite important, I suggest including, if possible, the amounts of the more abundant molecules detected.

NOT POSSIBLE FOR REASONS EXPLAINED IN 5 ABOVE.

Minor comments

Along the text I made some observations, please check it all.

THANK YOU!

ml IS CORRECTED THROUGHOUT

PHOSPHATE CONCENTRATION IS ALREADY PROVIDED IN TABLE 6 WHERE THE TIME COURSE OF THE OTHER COMPOUNDS IS ALSO SUMMARIZED!

LINE 239: COLUMN SPECIFICATION ADDED

LINE 241: REAGENT PURITY AND BRAND ADDED

TABLE 2 IS A SUMMARY. THIS POINT HAS BEEN ADDED TO THE CAPTION, AS HAS THE QUESTIONABLE ISSUE OF WHETHER THE PROLINE MATCH IS SUFFICIENTLY GOOD (SEE METHODS NOTE ABOUT QOM AS WELL).

LINE 401: “METABOLITES” CLARIFIED BY ADDING EXAMPLES.

LINE 462: EXPLAINED WHY OXYGEN MAY BE A PROBLEM

LINE 539 AND 574: DUPLICATE “FIGURE 39” NUMBERING CHECKED AND CORRECTED

TABLE 6: “602” ELIMINATED.  SUCCINIC ACID IS ANOTHER NAME FOR BUTANOIC ACID (THUS THE PARENTHESES) – THE CONFUSION AROSE FROM A COMMA BETWEEN THE TWO, WHICH SHOULD NOT BE THERE AND HAS BEEN REMOVED.

FIGURE 45 CAPTION TYPOS CORRECTED

ALL MODIFICATIONS OTHER THAN TYPOS ARE HIGHLIGHTED IN YELLOW IN THE REVISED MANUSCRIPT FOR EASE IN IDENTIFICATION.

WE THANK THE REVIEWER FOR THE CARE S/HE HAS TAKEN IN REVIEWING THE MANUSCRIPT!

Submission Date

20 December 2022

Date of this review

29 Dec 2022 00:51:29

Reviewer 3 Report (New Reviewer)

Interesting work, with great analytical scope. The idea of seawater application in Urey-Miller experiment  is unique, adding calcium phosphate and magnesium sulfate is the way how to synthesize more complex systems.

 After reading and assessing the work I have only a few formal comments. One  concerning the citation  of more recent published papers, concerning the detection of sugars in Miller type experiments ( for example, Civiš S, et al. (2016) TiO2-catalyzed synthesis of sugars from formaldehyde in extraterrestrial impacts on the early Earth. Sci Rep 6:23199) when TiO2 and its catalytic properties are used.

My second comment is related to  formal style of the publication. The publication is poorly arranged by the number of  individual  GC-MS Figures.

I recommend moving most of the GC-MC  identification figures into the  Supplementary information of the journal.

Author Response

REVIEWER 3

Open Review

English language and style

( ) English very difficult to understand/incomprehensible
( ) Extensive editing of English language and style required
( ) Moderate English changes required
(x) English language and style are fine/minor spell check required
( ) I don't feel qualified to judge about the English language and style

Yes

Can be improved

Must be improved

Not applicable

Does the introduction provide sufficient background and include all relevant references?

(x)

( )

( )

( )

Are all the cited references relevant to the research?

( )

(x)

( )

( )

Is the research design appropriate?

(x)

( )

( )

( )

Are the methods adequately described?

(x)

( )

( )

( )

Are the results clearly presented?

( )

(x)

( )

( )

Are the conclusions supported by the results?

(x)

( )

( )

( )

Comments and Suggestions for Authors

Interesting work, with great analytical scope. The idea of seawater application in Urey-Miller experiment  is unique, adding calcium phosphate and magnesium sulfate is the way how to synthesize more complex systems.

 After reading and assessing the work I have only a few formal comments. One  concerning the citation  of more recent published papers, concerning the detection of sugars in Miller type experiments ( for example, Civiš S, et al. (2016) TiO2-catalyzed synthesis of sugars from formaldehyde in extraterrestrial impacts on the early Earth. Sci Rep 6:23199) when TiO2 and its catalytic properties are used.

ADDED! (NEW REFERENCE 18)

My second comment is related to  formal style of the publication. The publication is poorly arranged by the number of  individual  GC-MS Figures. I recommend moving most of the GC-MC  identification figures into the  Supplementary information of the journal.

SHIFTED TO APPENDICES (WE PREFER APPENDICES TO SUPPLEMENTS IN ORDER TO KEEP THE DATA EASILY AVAILABLE IN THE ORIGINAL PUBLICATION RATHER THAN REQUIRING THE READER TO ACCESS SEPARATE MATERIAL, WHICH SOMETIMES FAILS TO LINK TO THE ORIGINAL ARTICLE AND DOES NOT DOWN-LOAD WITH THE PDF PUBLICATION).

 Submission Date

20 December 2022

Date of this review

07 Jan 2023 20:16:28

This manuscript is a resubmission of an earlier submission. The following is a list of the peer review reports and author responses from that submission.

Round 1

Reviewer 1 Report

The authors report a new analysis in this manuscript that replicates Miller's experiments. I find it generally very interesting and the results are fully described. However, I must make some essential observations in relation to what is described in the results, which must be taken into account very seriously by the authors. On line 122, type the phrase with the missing space between the words.

The surprise for me is the synthesis of the L-enantiomers (Fig. 6) of amino acids and mainly the D-enantiomers in sugars.

In another line, the authors write they have "LI- Glutamine"

What is the "LI-Glutamine" in Table 2?

Authors must also change

 the reference style on line 421 and review the structure of the text.

These details are important but are not considered relevant as the results where almost only the L-enantiomers are obtained in amino acids and the D-enantiomers in sugars. Unfortunately, this result is not reliable from the authors' methodology. In this sense, a control experiment is expected for instance by synthesizing the crystalline phases from “sea water” in the laboratory in comparison with the samples they have. Was an astecCHIROBIOTIC column at some point used to characterize the optically active molecules? This point is my most profound concern, since the synthesis of molecules of L amino acids and D sugars, implies a clear physicochemical bias. Thus, it is essential to point out that as long as the way to characterize the entire chiral organic composition of the presented synthesis is not described, this manuscript cannot be accepted. Examples of biased synthesis exist (e.gr. Soai) towards chirality and the authors here show almost a biased biosynthesis from inorganic compounds. In other words, I suggest the authors guarantee that L and D enantiomers are or are not present in the experimental results.

Reviewer 2 Report

Thank you for the opportunity to review this manuscript. To be honest, when I read the title I was completely surprised and I thought "well, maybe the authors solved the origins of life!" After a first peruse of the manuscript, which has not been easy due to the many flaws I found, I thought "is this maybe a fabricated manuscript oriented to check the quality of the review process?". 

I do not know where to begin. My first recommendation for authors it to look for advice from experts in the field before resubmit the manuscript. Second recommendation is to improve the quality of presentation. The low quality of figures, specially figures 3 and 6, is outrageous and cannot be accepted in any reputable journal.  

Regarding the results, the authors found exactly what is expected in a Miller-type experiment, and they should improve  the discussion of results. The (hard to read) molecules identified in figure 6 are consistent with what we know in this kind of experiment. It is surprising also that they just provide ONE chromatogram. They present the results in tables, without showing neither chromatograms nor mass spectra. This is not acceptable. 

A special mention should be done to the purported finding of 'nucleic acids'. First, they list as nucleic acid  the adenosine and free purines. This is misleading and questionable.

Second, have the authors considered a biological origin for their finding of minute amounts of adenosine, sugars and other compounds, like long chain fatty acids? I will give them a clue or two: stearic acid (C16) and arachidonic acid are usually markers of biological contamination, and its source is usually human (our finger's grease). Also, if adenosine is identified as just one peak or two peaks (depending on the quality of silanization) in GC-MS, you probably have a biological product. Mass spectral database identify as adenosine to several isomers when its origin is abiotic, because, unless you solved the problem of selection, you will form adenine nucleosides with several pentoses, and each one with two anomers with different reactivity to MSTFA. Another clue to biological origin:  the finding canonical, biological cAMP (3',5'-cAMP) and not 2',3'-cAMP, which is the preferred isomer in abiotic conditions. The presence of myo-inositol is another clue of contamination. Same could be said about ATP. So, you claim to form a high energy, unstable molecule, but not the 5',3',2'-ATP, or the 5'P-2',3'-cAMP? this is anothe clue to biological contamination. 

Well, overall, you have been deceived by the sensitivity of the analytical methods. You found the usual Miller-type experiment product, plus a suite of biological products. To claim what you claim in your manuscript would require a lot of work and proofs that you do not provide. Moreover, I strongly recommend again to look for advice before trying to publish that, for your own good. Also, I recommend to read more, as your mixture have serious chemical compatibility issues. The lack of sugars in Miller experiments has a chemical explanation, for example; also, there are no prebiotic synthesis of polyunsaturated fatty acids, as arachidonic, or eicosatrienoic acids). Your report of fatty acids is compatible with biological origin)

Sorry for being harsh, but, to be honest, I felt quite upset reading your manuscript.  If you want, I gladly offer my advice to revise and discuss your experimental results (specially because you have not presented it properly in the main manuscript, and I have not found any supplementary information) and assist you in the preparation a more proper work. 

Reviewer 3 Report

Report for the manuscript submitted to Life

-        “Sea Water” …

From Root-Bernstein and co-workers

The presented article is a new Miller experiment and the results are average described but poorly presented and however the manuscript deserve to be published in Life, matching with the aim of the special issue, need to be implemented in the following points

1.      Place in the Appendix as Life template reports ALL the GC-MS and LC-MS of the presented results and not only one or two over thousends that for sure were performed by the authors.

2.      The manuscript needs to be implemented in the format, whereas the authors did not followed authors instructions, see the word template in LIFE website for a better article presentation.

3.      The simple summary paragraph is not demanded and I suggest to erase

Title is very long. I suggest to cut down or modify it like “concomitant synthesis of amino-acids, Sugars, Nucleotides and lipids in a Miller-like experiment”

Line 70 – the reference Zaia 2008 have no number, please provide to the correction

FIGURE 1 does not contain any yields as reported at line 423, add or provide to modify or to adjust the text in function of what you need to report

Line 118 is 140° and not 140o, please correct

Line 122 add a . between deonized and The

Lines 139 to 143 : A few questions: why the concentration of the sea water is the half of the actual one? There is a geochemical reason? If yes please discuss and cite if possible any useful article. Apatite is not a very soluble phosphate salt. Why the use of this one and not another more soluble? Please justify your choice

Lines 179 to 195 : Mass spectrometry. I suggest to cite for the method the following article whereas the method you are describing is fully elucidated.

A new GC-MS method for the analysis of ascaulitoxin, its aglycone and 4-aminoproline from culture filtrates of Ascochita caulina

Michele Fiore, Agnes Rimando, Anna Andolfi, Antonio Evidente

Anal. Methods, 2010, 2, 159–163

Line 225 Results: The figure is not clear. Please erase Miller stopped here and place an * instead. In the legend you can then add the * and say that Miller have stopped in that point his sampling and so on. You can also erase gas reloaded adding another symbol like § and in the legend add gas reloaded. The text is clear about a gas reloading every week, however day 8 and 17 are a little bit more than 7 days as reported but if I’m not wrong is 9 days. Can you correct and be more clear? Like saying that the reloading was at days 8, 17 and so on?

This have to be corrected in the materials and methods and in the results and in the discussion as well.

Line 259 (FIGURE 4) in this figure you have reported one spectra of the glycine that seems to be identical. At page 8, first sentences the authors reported that several aminoacids were detected together with some “reactants” preferable to “reactant compounds”

Figure 5 and table 1 report this results. However as a scientific article, it is needed that in the Appendix A the GC-MS/LC spectra needs to be reported. Please follow the instruction for authors to complete this part. The same MUST be done for the analysis of amino acids, sugars and the other classes of presented compounds : sections 3.3 to 3.6

Pages 9, 11,12-13,14, 18 and 20 : all the tables need to be formatted in MPDI Life style

Line 280: Figure 6 is not acceptable in the present format. Plese provide a suitable high quality JPEG or TIFF image. It seems to be a print screen.

Figure 7 please correct in methionine and cysteine

Figure 8 sugars MUST be present without any Cram notation? The authors have not mentioned any region and stereo selection for this class of molecules and reasonably they are present as a racemic mixtures

Figure 9, same as for FIGURE 8

Table 3 : the spectra relative to the analysis of the sugars MUST be placed in Appendix part.

Table 4 : same comment for table 3

Line 350 cGMP present one extra oxygen, please provide to a correction in Figure 11,  

Discussion and Conslusions are placed in the same paragraph, where end the discussion and where start conclusions? I suggest to revise these two parts

Line 448 is instead if Is

General comment on the manuscript

Implementation of the Miller experiments after the analysis of his forgotten samples is of great interest in the field of origin of life studies. However this article is not presented as the results obtained deserve to be presented.

This referee can’t contest the presence of the large biotic relevant molecules described into the text. Nowadays, chemists as well as biologist cannot tell a story without presenting the results in a clear manner. Figures have to be accompanied by a large text and then in Appendix A, B, C and so on supplementary figures must be placed, together with additional discussions to complete and support the results presented in the main manuscript.

I warmly invite the authors to present all the results in a more clear manner eliminating any doubt on the experiments done AND analysis done.